METHODS

# A surface morphology-based inference method for the cell wall elasticity profile in tip-growing cells

Rholee Xu[1], Luis Vidali[1,2]*, Min Wu[1,3]*

**1** Bioinformatics and Computational Biology Program, Worcester Polytechnic Institute, Worcester, Massachusetts, United States of America, **2** Department of Biology and Biotechnology, Worcester Polytechnic Institute, Worcester, Massachusetts, United States of America, **3** Department of Mathematical Sciences, Worcester Polytechnic Institute, Worcester, Massachusetts, United States of America

* lvidali@wpi.edu (LV); englier@gmail.com (MW)

## Abstract

Plant development and adaptation are highly dependent on cell morphology and growth. High turgor pressure in plants causes stress on the cell wall, followed by cell extension. In tip-growing cells, the localization of vesicles and cytoskeleton components has been well studied. However, there has been a lack of attention to the spatial profile of mechanical properties, specifically the cell wall elasticity. In this study, we introduce a new surface morphology-based method to measure the elasticity of the cell wall in tip-growing cells. Previous work is based on measurements from the wall meridional outline, a technique that cannot track the elastic deformation of the cell wall experimentally. Instead, we developed a way to infer the bulk modulus distribution from the cell surface by triangulating experimental marker points coming from fluorescent labeling. To justify the use of our protocol in tip-growing cells from the moss *Physcomitrium patens*, we replicated the experimental noise and moss morphology in simulated cells. In practice, we found that a larger triangulation improved robustness against noise, which agreed with our theoretical study. With multiple cell sampling, we determined that 10 cells were sufficient to recover the elasticity distribution with noise, but only when the elastic stretches were high enough. We then created a dimensionless map of inference error to verify a spatial change of *P. patens* bulk modulus within two folds. This technique will open the field to more comprehensive measurements of cell wall elasticity, providing a key step in understanding tip cell growth and morphogenesis.

## Author summary

Tip-growing cells can be characterized by their fast growth concentrated at the cell's apex. Their growth and morphogenesis are tightly regulated processes involving cell wall addition and rearrangement while the cell wall is under stress

**Data availability statement:** All relevant data and code, including code demonstrations, are available on the GitHub repository found here: https://github.com/rholee-xu/surface-model.

**Funding:** National Science Foundation - Division of Mathematical Sciences (https://www.nsf.gov/mps/dms) funded DMS-2012330 and DMS-2144372 (CAREER) grants to M.W. National Science Foundation - Division of Molecular and Cellular Biology (https://www.nsf.gov/bio/mcb) funded MCB-2154573 grant to L.V. The funders had no role in study design, data collection and analysis, decision to publish, or preparation of the manuscript.

**Competing interests:** The authors have declared that no competing interests exist.

originating from the cell's internal turgor pressure. We start by studying the cell wall's elastic properties, one aspect of the cell growth process. We use a method of marker point tracking across the surface of the tip-growing cell to measure the wall's elasticity profile. In this work, we present a parameter sensitivity study of this method on synthetic cells and report our results on experimental moss tip-growing cells. Our results suggest that this inference method can reliably measure a cell wall elasticity gradient under combined geometric and mechanical conditions that create elastic strains within 5% at the tip.

## Introduction

The dynamic ability of cell walls to consistently add and rearrange wall material while extending is often understated, as we commonly view the wall as a rigid material. The wall itself defines the morphology of walled cells and provides protection, but must sustain internal pressures up to 1000 times higher than in animal cells [1–3]. Hence, the morphogenesis and growth of walled cells strongly depend on the mechanics of the cell wall, especially how those mechanical properties differ in particular areas along the cell surface. Tip-growing cells are an ideal system for studying these spatial differences because of their simple morphology and concentrated surface expansion at the tip of the cell. Given that this feature is conserved across many eukaryotic species, it is likely to have been selected to support fast linear growth [4]. In plant species, tip-growing cells such as root hairs are essential for acquiring water and nutrients [5], and pollen tubes are needed for fertilization [6]. Additionally, moss protonema like in the model organism *Physcomitrium patens*, utilize tip growth for fast surface area expansion [7]. Outside of plants, tip-growing cells like fungal hyphae or water molds may serve other functions such as invasive growth, but with a similar underlying goal of efficient expansion [4,8,9]. Understanding how tip-growing cells can maintain stable cell wall growth against their strong internal turgor pressure is essential to the field of plant growth and morphology.

The small scale of single tip-growing cells make it difficult to measure the mechanical properties of the wall, especially at discrete spatial locations. These measurements have been more easily accomplished at larger scales, for instance in plant tissues. The physical clamping and stretching of onion epidermal cells is one such example [10–12]. In tip-growing cells, the micro-indentation technique has been used to measure the spatial variation in pollen tube cell wall mechanical properties [13–15]. However, due to the geometrical orientation of the probing, the measurement of the mechanical properties may not be as relevant to wall expansion. Instead of experiments, many mechanical models have been developed since the Lockhart model, which describes wall extension as a function of wall extensibility, regulated by turgor pressure [16]. Subsequent tip growth models have expanded upon this idea by adding a spatial component, while taking advantage of the cell axial symmetry [9,17–22]. Results from models indicate that a steep gradient of mechanical properties moving away from the tip is needed to produce efficient cell wall extension that

matches observed growth and realistic morphologies. A more detailed depiction of the Lockhart model describes the wall growth coming from the combination of two actions, the first of which is the reversible elastic deformation defining the inflation of a cell without turgor pressure to a pressurized configuration [23,24]. The second action involved is the irreversible wall expansion, coming from wall biochemical processes. The connection between these two processes was explored in yeast, where they first found cell wall elastic changes of up to 20% when the turgor pressure was removed [23]. When the morphogenesis was modeled with wall elasticity, there was good agreement between the simulated wall extension and experiments [23,25,26]. More recently in filamentous fungi, similar plasmolysis experiments were used to estimate the elastic properties at the tip and shank [27]. Importantly, the results showed a significant difference between the cell side and tip's wall Young's Modulus. However, without the tracking of material points on the wall, the measurement assumes that the wall strain has the same trend as the wall curvature, which was not tested. To truly support the applicability of the second type of model in tip-growing cells, a more detailed measurement of the elastic properties at the region of transition from the side to the tip is needed.

The feasibility of an axisymmetric marker point inference scheme on the wall outline was demonstrated in [28] and [29] to track the wall tensions and elastic moduli, respectively. In experiments, such marker point tracking can come from fluorescent beads (Fig 1A) or quantum dots that can be adhered to the cell wall surface [23]. In practice, we found that we could not accurately track the experimental fluorescent beads along the meridian of a single wall outline, due to general rotational movement after deformation. Instead, we developed a surface morphology-based method pipeline to measure the deformation directly on the wall surface. The method will compute the surface curvatures and elastic deformations, and then reduce them into the elastic moduli in the axisymmetric wall outline model. In particular, we generated simulated cells from experimental *P. patens* cell wall outlines to evaluate the method with varying levels of noise estimated from experiments. Our results suggest that, when there is noise, the elasticity profile can be recovered by using a coarser discretization of triangles and multiple cell samples. Ultimately, we created a dimensionless reference map that can be used to verify the inference in experimental cells, including *P. patens*.

## Methods

### Cell wall surface tracking of elastic deformation and curvature

**Marker point localization and automated triangulation:** To triangulate the surface, we first selected a subset of marker points around the cell surface (Fig 1A and 1C). For the marker points, we used fluorescent microspheres adhered to the cell wall non-specifically (Fig 1A and S1 Text). To ensure the accuracy of the marker points, we localized them to a sub-pixel resolution using the software RS-FISH [30], a fluorescent spot detector applied in ImageJ (S1A Fig). We plasmolyze the cell with an osmoticum change to obtain the cell configuration with no turgor pressure (Fig 1A bottom). Moving forward, we refer to the plasmolyzed cell as the unturgid configuration, and the pressurized cell as the turgid configuration. The matching of marker points between the two configurations is done manually by scatter plotting the two sets and visually pairing them (S1C Fig and Sect 1.2 in S1 Text).

To standardize our triangulation process, we devised an automated method by Delaunay triangulation (Fig 1D). We employ the `delaunay` function in Matlab [31]. We found that a 3D Delaunay did not ensure the angle quality of the triangulation well (S2 Fig). Therefore, we used a 2D Delaunay triangulation by first applying a stereographic-like projection to the points onto an x-y-plane above the cell (Fig 1D, i–iii). Due to the shape transition of the cell, the marker points at the tip are projected from a single point located at the center of that area, while in the pipe region each marker point is projected from a point on a center line through the cell (Fig 1D, ii). After projection, we applied the delaunay function to obtain our connectivity map (Fig 1D, iv). The same connectivity map is applied to the original marker points to obtain the final triangulated surface (Fig 1E). Given the one-to-one correspondence of marker points in the turgid and unturgid configuration, the same triangulation is used for the unturgid cell.

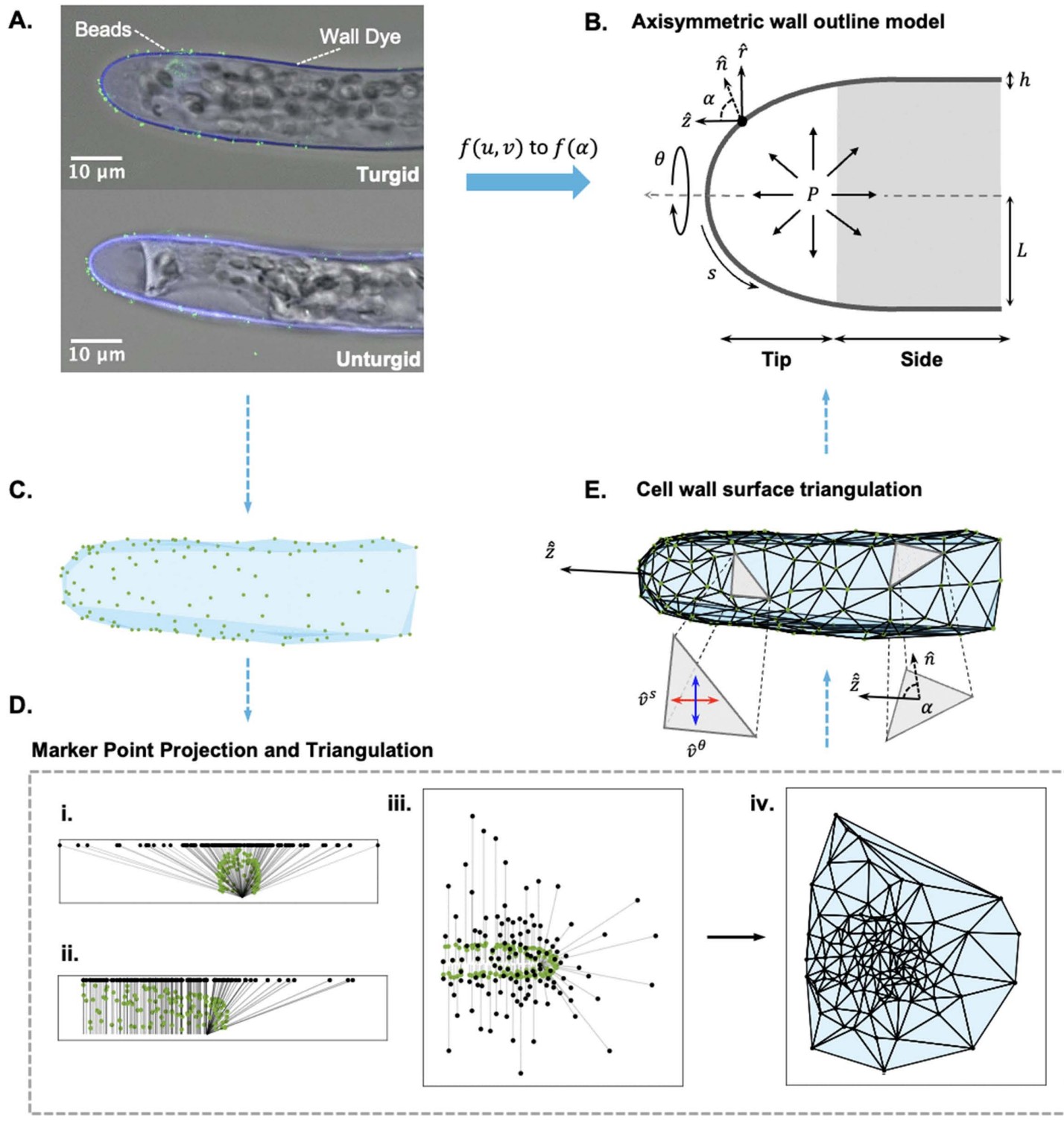

**Fig 1. Marker point triangulation scheme connects the cell wall surface to the axisymmetric wall outline. A.** *P. patens* apical cell labeled with calcofluor white and adhered with fluorescent beads (Thermofisher). The unturgid cell (below) comes from the plasmolysis of the turgid cell (above) after the addition of a 15% mannitol solution. **B.** Standard thin-shell axisymmetric outline model of a tip growing cell. Non-dimensional parameters include the turgor pressure ($P$), cell wall thickness ($h$), cell radius ($L$), and the circumferential ($\theta$) and meridional ($s$) coordinates (see Methods). Each wall point

has a local angle ($\alpha = \cos^{-1}(\hat{n} \cdot \hat{z})$) between the outward normal ($\hat{n}$) and long axis ($\hat{z}$). The gray area denotes our demarcation of the cell side from the cell tip area (which covers a range of $80 \leq \alpha \leq 90$). **C.** Translation of the fluorescent beads into Matlab. **D.** Automated bead projection and triangulation process. **(i-iii)** Bead projection onto x-y-plane from three views: **(i)** front, **(ii)** side, and **(iii)** top. **(iv)** Automated triangulation in the 2D plane. **E.** Final triangulation on the 3D cell. Different components include the new optimized long axis, $\hat{\tilde{z}}$, the calculation of $\alpha$ on each triangle (S3 Fig), and the optimized circumferential and meridional directions, $\hat{v}_s$ and $\hat{v}_\theta$, respectively (see Methods).

**Computation of the elastic deformations from the wall surface:** In finite strain theory, the deformation gradient relates a reference and current configuration of a given region [32]. In our case, we use the deformation gradient, **F**, to connect the surface triangulation patches in the turgid and unturgid configuration (Fig 2A). The pair of triangles are defined by $\mathbf{B} = [\vec{v}_1^b, \vec{v}_2^b, \hat{n}^b]$ and $\mathbf{A} = [\vec{v}_1^a, \vec{v}_2^a, \hat{n}^a]$, respectively. Here, $\vec{v}_1$ and $\vec{v}_2$ are the vectors defining any two sides of

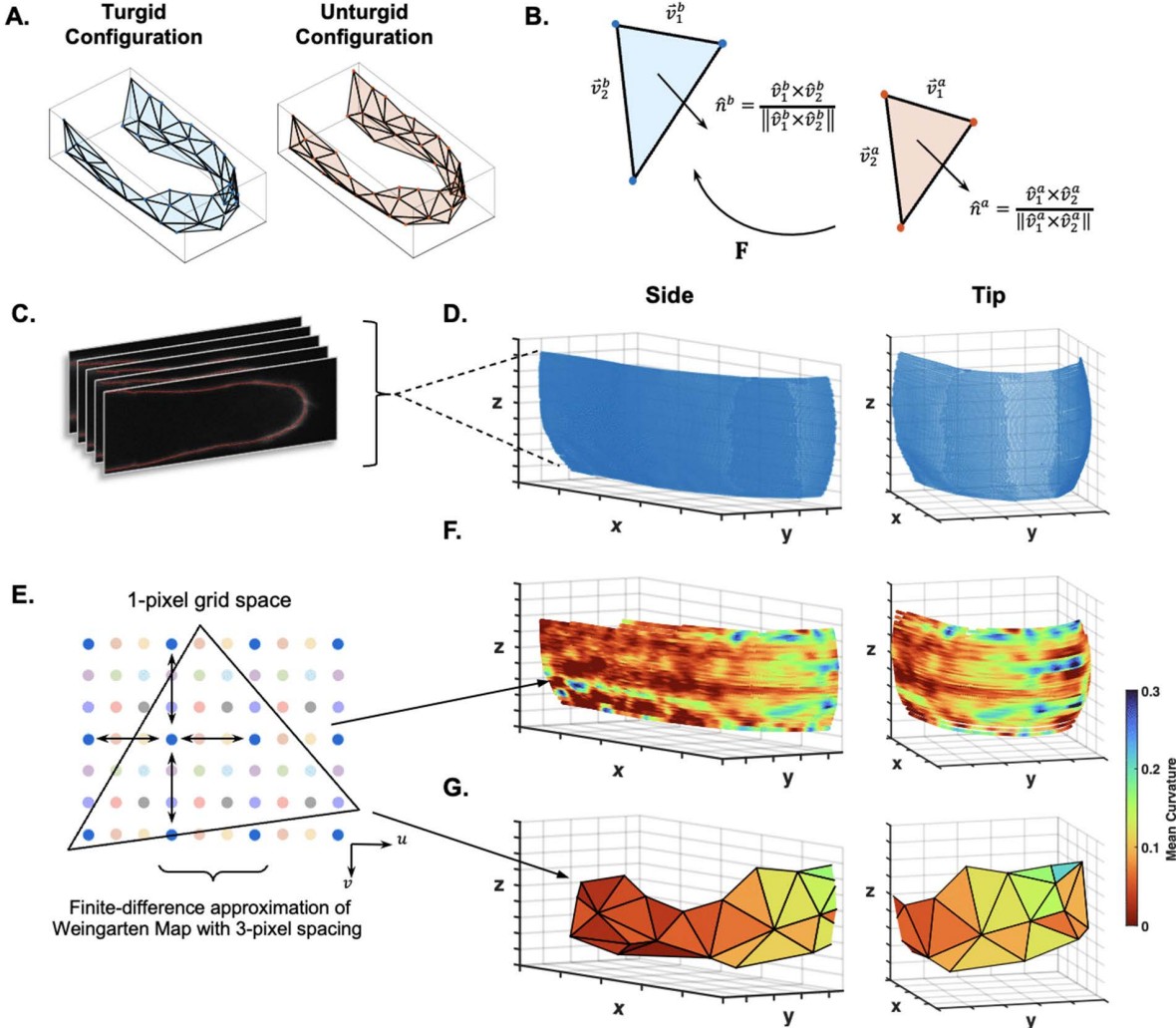

**Fig 2. Elastic stretch and curvature calculation on the triangulated cell wall surface. A.** Turgid and unturgid triangulations (following Fig 1). **B.** The linear map **F** describes the deformation of the unturgid triangle (red) to the turgid triangle (blue) through each triangle's side vectors ($\vec{v}$) and normal vectors ($\hat{n}$). **C.** Collection of Z-stack images collated to reconstruct cell wall surface. **D.** Raw data of an experimental cell surface, shown from the side and tip. **E.** Demonstration of the finite-difference approximation with a 3-pixel spacing. Shifting covers the entire 1-pixel grid space. **F.** Mean curvature values on each wall point associated to each triangle are averaged **(G)**. See S4 Fig for more details.

the triangle, and $\hat{n}$ is the triangle's normal vector (Fig 2B). The polar decomposition of the deformation gradient **F** is: **F** = $\lambda_1 \hat{x}_1^b \otimes \hat{x}_1^a + \lambda_2 \hat{x}_2^b \otimes \hat{x}_2^a + \hat{n}^b \otimes \hat{n}^a$, where $\lambda_1$ and $\lambda_2$ are the principal stretch ratios in the local tangent plane. Notice that $\hat{n}^b \otimes \hat{n}^a$ describes the rotation of the local tangents. The right and left Cauchy-Green tensors are $\mathbf{F}^T\mathbf{F}$ and $\mathbf{F}\mathbf{F}^T$, respectively [33]. The square root of the non-unity eigenvalues of these matrices gives the values of $\lambda_1$ and $\lambda_2$. The eigenvectors of $\mathbf{F}^T\mathbf{F}$ that are not $\hat{n}^a$, describe the principal directions in the unturgid configuration, $\hat{x}_1^a, \hat{x}_2^a$. The eigenvectors of $\mathbf{F}\mathbf{F}^T$ that are not $\hat{n}^b$, describe them in the turgid configuration $\hat{x}_1^b, \hat{x}_2^b$. In summary, the decomposition of **F** shows that **F** is simply a way to transform any vector on the tangent plane of the unturgid patch to the turgid patch, including the unit normal vectors (Fig 2B). Conversely, we can use $\mathbf{F}^{-1} = \frac{1}{\lambda_1}\hat{x}_1^a \otimes \hat{x}_1^b + \frac{1}{\lambda_2}\hat{x}_2^a \otimes \hat{x}_2^b + \hat{n}^a \otimes \hat{n}^b$ to transform a vector on the turgid patch to the unturgid patch. The calculation of $F$ in practice can follow: $\mathbf{F} \cdot \mathbf{A} = \mathbf{B}$. Therefore:

$$\mathbf{F} = \mathbf{B} \cdot \mathbf{A}^{-1} \tag{1}$$

The linear map **F** and the principal directions, $\hat{x}_1^b, \hat{x}_2^b$, are used as a local basis to find the optimized circumferential and meridional directions in a later section.

Since **A** and **B** are constructed from the coordinates of triangle vertices, we analyzed how perturbations on the triangle vertices are propagated to the calculation of **F** (see Sect 1.4 in S1 Text). In particular, we derived a formula $\frac{\|\delta\mathbf{F}\|_2}{\|\mathbf{F}\|_2} \leq \Gamma|\delta|$, where $\frac{\|\delta\mathbf{F}\|_2}{\|\mathbf{F}\|_2}$ is the relative error, and $\Gamma|\delta|$ is a function of geometric parameters related to the triangle. The $|\delta|$ term sets the scale for some perturbation on the vertices (S1 Text). Although the $\Gamma$ function does not appear insightful analytically due to the complexity of the function, it can be used to predict the sensitivity of the deformation gradient to noise. It acts as an upper bound of error a triangle may potentially have, given a level of noise, $|\delta|$. We used the distribution of triangle $\Gamma$ values to set a threshold to remove more sensitive triangles. We found that this effectively retained triangles with a more compact shape. Once we define the magnitude of noise, $|\delta|$ (see "Noise on the marker points" section), we will report more on this triangle analysis.

**Computation of curvatures from the wall surface:** We extracted the curvature tensors from the surface data in order to make the curvature and subsequent tension calculations compatible with the elastic stretch ratios. The detailed description of the curvature procedure can be found in Sect 1.3 in S1 Text. In short, we first used the Ridge Detection software in ImageJ [34,35] to extract the available cell wall outline data (Fig 2C and S1B Fig). Then we used the point cloud data to parameterize the surface regions depending on the surface orientation (Fig 2C and 2D). Specifically, we used the wall outline of the plasmolyzed cell. The resulting grid parameterization, $X(u,v) = (x,y,z)$, is used to calculate the Weingarten Map, $W$ [36]. The eigenvalues of $W$ give the principal curvatures, $\bar{\kappa}_1(u,v)$ and $\bar{\kappa}_2(u,v)$ (Fig 2E and 2F). The effect of the finite difference discretization is also described in Sect 1.3 in S1 Text and seen in S4 Fig. The results of this analysis are summarized in (Fig 2E, 2F).

## Cell wall mechanics model

**Axisymmetric cell wall outline model:** To model the mechanics of the cell on the wall outline, we describe the wall as a thin pressurized shell revolved around an axis of symmetry (Fig 1B). In the axisymmetric system, the wall properties are defined along two orthogonal directions, the circumferential ($\theta$) and meridional ($s$) directions. All properties can then be expressed as a function of the meridional coordinates, $s$. The cell wall stresses in the circumferential and meridional directions ($\bar{\bar{\sigma}}_\theta, \bar{\bar{\sigma}}_s$) must balance the force in the system coming from the internal turgor pressure, $\bar{P}$. This results in the Young-Laplace Law which can be used to infer the wall tensions from the wall curvatures ($\bar{\kappa}_\theta, \bar{\kappa}_s$) [17,18,28,37].

$$\bar{\sigma}_s = \bar{P}/(2\bar{\kappa}_\theta) \tag{2}$$

$$\bar{\sigma}_\theta = \bar{P}/(2\bar{\kappa}_\theta) \times (2 - \bar{\kappa}_s/\bar{\kappa}_\theta) \tag{3}$$

Here, the wall tensions, $\bar{\sigma}_s$ and $\bar{\sigma}_\theta$, are equal to in-plane wall stresses multiplied by the cell wall thickness $h$: $\bar{\sigma}_\theta = \bar{\bar{\sigma}}_\theta \times h$ and $\bar{\sigma}_s = \bar{\bar{\sigma}}_s \times h$. To describe the cell wall mechanics, we assume that the tensions are sustained by the cell wall elasticity.

While many cell walls may present anisotropic material property due to the fiber alignments [2], there are cell walls such as moss [38] that have fibers aligning with the wall surface, and distribute without preferred directions within the local plane. This can justify a 2D isotropic material [17]. The resulting constitutive law describes how the tensions change in response to the elastic stretch ratios, modulated by the two material properties, the surface bulk and shear modulus, $\bar{K}_h$ and $\bar{\mu}_h$, respectively [28,29].

$$\bar{\sigma}_s = \frac{1}{2}\bar{\mu}_h\left(\frac{1}{\lambda_\theta^2} - \frac{1}{\lambda_s^2}\right) + \bar{K}_h(\lambda_s\lambda_\theta - 1)$$

(4)

$$\sigma_\theta = \frac{1}{2}\bar{\mu}_h\left(\frac{1}{\lambda_s^2} - \frac{1}{\lambda_\theta^2}\right) + \bar{K}_h(\lambda_s\lambda_\theta - 1)$$

(5)

The elastic stretch ratios, $\lambda_s$ and $\lambda_\theta$, describe the change from the unturgid to turgid configuration. These can be related to the elastic strains, $\epsilon$ as follows: $\epsilon_s = \lambda_s - 1$ and $\epsilon_\theta = \lambda_\theta - 1$. When the elastic strains are small, $|\epsilon| << 1$, we observe that our nonlinear constitutive law is equivalent to a linear elastic law [24]. In this circumstance, the surface bulk and shear modulus can be connected to the surface Young's modulus and Poisson's ratio through: $\bar{\mu}_h = \frac{\bar{E}_h}{2(1+\nu)}$ and $\bar{K}_h = \frac{\bar{E}_h}{2(1-\nu)}$. The surface moduli can all be related to the 3D moduli through $h$: $\bar{E}_h = \bar{E} \times h$, $\bar{K}_h = \bar{K} \times h$ and $\bar{\mu}_h = \bar{\mu} \times h$. When there is enough information on the distributions of the wall thickness, $h$, this part of the equation can be incorporated.

Given the turgid and unturgid configurations of a cell, the elastic moduli $\bar{K}_h$ and $\bar{\mu}_h$, can be inferred from the elastic stretch ratios and tensions [29]. This inference can be seen by rearranging Eqs 4 and 5:

$$\bar{K}_h = \frac{\bar{\sigma}_s + \bar{\sigma}_\theta}{2(\lambda_s\lambda_\theta - 1)} = \frac{P(3 - \kappa_s/\kappa_\theta)}{4\kappa_\theta(\lambda_s\lambda_\theta - 1)}$$

(6)

$$\bar{\mu}_h = \frac{\bar{\sigma}_s - \bar{\sigma}_\theta}{1/\lambda_\theta^2 - 1/\lambda_s^2} = \frac{P(1 - \kappa_s/\kappa_\theta)}{2\kappa_\theta(1/\lambda_\theta^2 - 1/\lambda_s^2)}$$

(7)

On the right side of the equation, we have written $\bar{K}_h$ and $\bar{\mu}_h$ as a function of their geometrical descriptors using Eqs 2 and 3. In the results, we will present the non-dimensionalized variables: $\kappa_{s,\theta} \rightarrow \bar{\kappa}_{s,\theta}\bar{L}$, $\sigma_{s,\theta} \rightarrow \bar{\sigma}_{s,\theta}/\bar{P}\bar{L}$, and $K_h \rightarrow \bar{K}_h/\bar{P}\bar{L}$.

**Instability of the bulk and shear modulus against small perturbations:** Previously in [29], the effect of perturbations on the tensions and stretches on $\bar{K}_h$ and $\bar{\mu}_h$ was analyzed. Similarly, we can derive from Eq 6 that the perturbation on $\bar{K}_h$ is:

$$\frac{\delta\bar{K}_h}{\bar{K}_h} = \frac{\delta(\sigma_s + \sigma_\theta)}{\sigma_s + \sigma_\theta} - \frac{\delta(\lambda_s\lambda_\theta)}{2(\lambda_s\lambda_\theta - 1)}.$$

(8)

In practice, we have observed that the perturbations on $\sigma_{s,\theta}$ are insignificant compared to the perturbations on $\lambda_{s,\theta}$ (see Section "Noise on the marker points" below). Therefore, $\frac{\delta\bar{K}_h}{\bar{K}_h} \approx -\frac{\delta(\lambda_s\lambda_\theta)}{2(\lambda_s\lambda_\theta - 1)}$, specifying how the error on the area stretch ratio, $\lambda_s\lambda_\theta$, affects the error on $K_h$. Subsequently, the inference of $\bar{K}_h$ will suffer when $\lambda_s\lambda_\theta$ approaches 1. When $\lambda_s\lambda_\theta < 1$, negative $\bar{K}_h$ values will appear. Since these values are non-physical, we remove calculations with negative $\bar{K}_h$ from all of our results.

On the other hand, the inference of $\bar{\mu}_h$ is unstable when $\sigma_s \approx \sigma_\theta$ and/or $\lambda_s \approx \lambda_\theta$ (Eq 7). Due to the isotropic nature at the tip of the cell, this instability will always occur there. Even in the purely axisymmetric case, we have already observed that the inference of $\bar{\mu}_h$ is sensitive at the tip [29]. For those reasons, this paper will focus on the robustness of inferring just the bulk modulus. Similar to other work, we will only measure one elastic property and make a generalization of $\bar{\mu}_h$ to support our analysis [4,27].

**Conversion of variables from the wall surface to the outline model**

**Computation of surface parameters in the circumferential and meridional directions:** To use Eqs 6 and 7 to infer the elastic moduli, we need to calculate the elastic stretch ratios and tensions in the circumferential and meridional directions. Therefore, the cell needs to be approximately axisymmetric so that these directions can be found in an optimized sense on each triangulated surface patch (Fig 1B $\theta$ and $s$, to Fig 1E blue and red arrows). We first define a vector $\hat{\tau}$ for each turgid configuration triangle that describes the direction from the center of the triangle to the tip position of the cell. The tip position is determined by the point with the highest x-value along the length of the cell (S1D Fig). Additionally, we define any direction in the local tangent plane of the triangle as: $\hat{v}(\phi) = \cos(\phi) * \hat{x}_1^b + \sin(\phi) * \hat{x}_2^b$, where $\{\hat{x}_1^b, \hat{x}_2^b\}$ is a local basis on the turgid triangle, and $\phi$ is an angle between $\hat{v}$ and $\hat{x}_1^b$. In particular, we use the local principal stretch directions as the local basis. The circumferential direction is then defined as the vector most perpendicular to $\hat{\tau}$: $\hat{v}(\phi^*) \cdot \hat{\tau} = 0$. Then we denote $\hat{v}^\theta = \hat{v}(\phi^*)$. The meridional direction is the vector most aligned with $\hat{\tau}$ and is equal to $\hat{v}^s = \hat{v}(\phi^* \pm \pi/2)$. This process is repeated to establish $\hat{v}^s$ and $\hat{v}^\theta$ on the tangent plane of each triangle (Fig 1E, $\hat{v}^s$ and $\hat{v}^\theta$ arrows).

The circumferential and meridional directions in the turgid configuration can be used to calculate the corresponding stretch ratios, $\lambda_\theta$ and $\lambda_s$. Since the optimized directions live on the turgid plane, we use $\mathbf{F}^{-1}$ to transform them to the unturgid plane. Therefore, the ratio of length change before and after the deformation along $\hat{v}^\theta$ and $\hat{v}^s$ is the magnitude of the original turgid vector over the magnitude of the transformed unturgid vector:

$$\lambda_s = \|\hat{v}^s\|/\|\mathbf{F}^{-1} \cdot \hat{v}^s\| = 1/\|\mathbf{F}^{-1} \cdot \hat{v}^s\| \qquad \lambda_\theta = \|\hat{v}^\theta\|/\|\mathbf{F}^{-1} \cdot \hat{v}^\theta\| = 1/\|\mathbf{F}^{-1} \cdot \hat{v}^\theta\| \tag{9}$$

To find $\bar{\kappa}_s(\alpha)$ and $\bar{\kappa}_\theta(\alpha)$ on the wall surface, we use the principal directions on each wall outline point instead of triangle, and the principal curvature directions will be used as the local basis. From there, we define $\psi$ as the angle between a principal direction associated with curvature $\bar{\kappa}_1$ and the local meridional direction, $\hat{v}^s$. Using Euler's equation [39], the meridional curvature is given by:

$$\bar{\kappa}_s = \bar{\kappa}_1 \cos^2 \psi + \bar{\kappa}_2 \sin^2 \psi \qquad \bar{\kappa}_\theta = \bar{\kappa}_2 \cos^2 \psi + \bar{\kappa}_1 \sin^2 \psi \tag{10}$$

The final calculation of $\bar{\kappa}_s(\alpha)$ and $\bar{\kappa}_\theta(\alpha)$ on each triangle comes from the average values of each triangle's corresponding points (Fig 2E triangle to Fig 2G).

**Computation of mechanical properties on the wall surface:** With the curvatures on each wall outline point, we used Eqs 2 and 3 to calculate the dimensional wall tensions, $\bar{\sigma}_s$ and $\bar{\sigma}_\theta$. The final tensions on the triangle also come from averaging the point-wise tensions. Along with $\lambda_\theta$ and $\lambda_s$, we can now use Eq 6 to calculate the dimensional bulk modulus, $\bar{K}_h$. There is also an alternative to Eq 6 to calculate the bulk modulus that does not require the deformation gradient. From Eq 6, we can see the term $\lambda_s \lambda_\theta$, which describes the area change of a triangle between the two configurations. This is approximately equivalent to $S_B/S_A$, where $S_B$ and $S_A$ are the triangle areas in the turgid and unturgid, respectively. The alternative calculation of $\bar{K}_h$ uses the triangulations of the two configurations, and then $S_B/S_A$ instead of $\lambda_s \lambda_\theta$.

Lastly, we redefine a location marker compatible with the triangulation. We use the local angle $\alpha$ as the positioning coordinate such that different cell sizes can be considered and combined. To calculate $\alpha$, we first orient the cell towards a central long axis, $\hat{z}$. Depending on the orientation of the cell in the acquired image, a new long axis, $\hat{\tilde{z}}$ (see Fig 1E $\hat{\tilde{z}}$

arrow), needs to be found in an optimal sense. Then we use $\hat{\bar{z}}$, and each wall outline point's outward normal vector, $\hat{n}$, to calculate a local angle $\alpha$ with the equation $\alpha = \cos^{-1}(\hat{n} \cdot \hat{\bar{z}})$ (Fig 1E, bottom right triangle). Since each triangle covers multiple wall points, we assign a range of $\alpha$ values to each triangle (S3 Fig). This range is then used to bin any parameter results for each triangle (see Sect 1.7 in S1 Text). Through these conversions, the related properties are transformed from $f(u,v)$ to $f(\alpha)$ (Fig 1 workflow).

### Simulated cell framework and estimation of procedure noise

**Simulated *P. patens* cell wall surfaces:** We adapted upon the previous inference methods using a synthetic cell outline [28,29], by generating two types of 3-dimensional simulated cells. To explore different morphologies, we have made use of two types of *P. patens* tip-growing cells. The caulonema cell is more tapered, and the chloronema cells are more round [40]. We have found that their canonical shapes match well with a hyphoid shape which can be described by the equation:

$$x = \frac{\pi y}{a} \cot(\pi y) - \frac{1}{a}$$

(11)

where $a$ is the parameter controlling the tapered nature of the cell [41]. Through the fitting of multiple experimental cell outlines, we determined the $a$ values for the two cell types (Figs 3A and S5).

We input $\bar{K}_h$ and $\bar{\mu}_h$ on the unturgid hyphoid outline with a range of distributions (see Sect 1.6 in S1 Text). Our computational model will solve for the corresponding turgid outline. To recreate the wall outline, we discretized the hyphoid outlines into 129 points. We additionally, discretized the hyphoid outlines into 17 points to create the set of initial marker points to select from. Both sets of points are then rotated about the long axis, thereby constructing the configurations of the three-dimensional cell wall surfaces (Fig 3B, i, iii). For each inference, we apply our automated triangulation scheme (Fig 1D) onto a subset of marker points (Fig 3C, i). The initial number of random marker points chosen ($N_\triangle$), and the distance boundary used to filter points too close to each another ($d_\triangle$) are adjustable. This will affect the resulting triangulation size. For a unique triangulation, the mean triangle area, $\bar{\triangle}_A$ can be non-dimensionalized using the cell radius $\bar{L}$: $\bar{\triangle}_A/\bar{L}^2$. Cells are given a random size within a range that covers variation observed in reality (Fig 3C, ii).

**Local elastic stretch ratio using change in radii:** To apply the procedure described in [27] to measure the wall surface modulus, $Y$, we used our turgid and unturgid hyphoid cell outlines. To match the parameters of their wall model, we set Poisson's ratio ($\nu$) equal to 0. Then, in relation with the bulk modulus, $2K_h = Y$. We generated caulonema and chloronema cells with this condition and three cases of elastic moduli distributions: constant, nonlinear, and linear. For the cell side, we used the formula: $Y_{side} = \frac{R_B}{(R_B - R_A)/R_A}$ where $R_B$ and $R_A$ are the radii of the cell side in the turgid and unturgid configurations, respectively. At the tip, we used the formula: $Y_{tip} = \frac{R_B}{2(R_B - R_A)/R_A}$. In both instances, we have used a rescaling of the formulas from [27] to remove the pressure ($\bar{P}$), and cell wall thickness ($h$). Given that it was not reported how tip radius was measured, we used a range of degrees of coverage of the wall outline. For each range, we used a sphere-fitting function to find the best fit sphere for the turgid and unturgid cells [42] (S7 Fig).

**Noise on the marker points and cell wall outline:** The Ridge Detection software used to extract the cell wall outline is optimized for finding the center of a fluorescent signal, as well as maintaining the smoothness of the curve [34,35]. Therefore, we attributed the noise to the software's ability to detect the real cell wall given the quality of fluorescence signal. To recreate this, we have added noise to the outline data by moving the wall surface outlines outwardly or inwardly along the normal direction (Fig 3B, ii).

The wall outline noise is independent of the marker point noise, one aspect of which comes from the RS-FISH software we used [30]. From their work, we have estimated that their accuracy is within 1% of the cell width of our synthetic cells, with larger error in the z-direction. This noise would be applied to the marker points before and after deformation. The larger aspect of marker point noise comes from the imaging setup itself, including the disturbances in the experimental

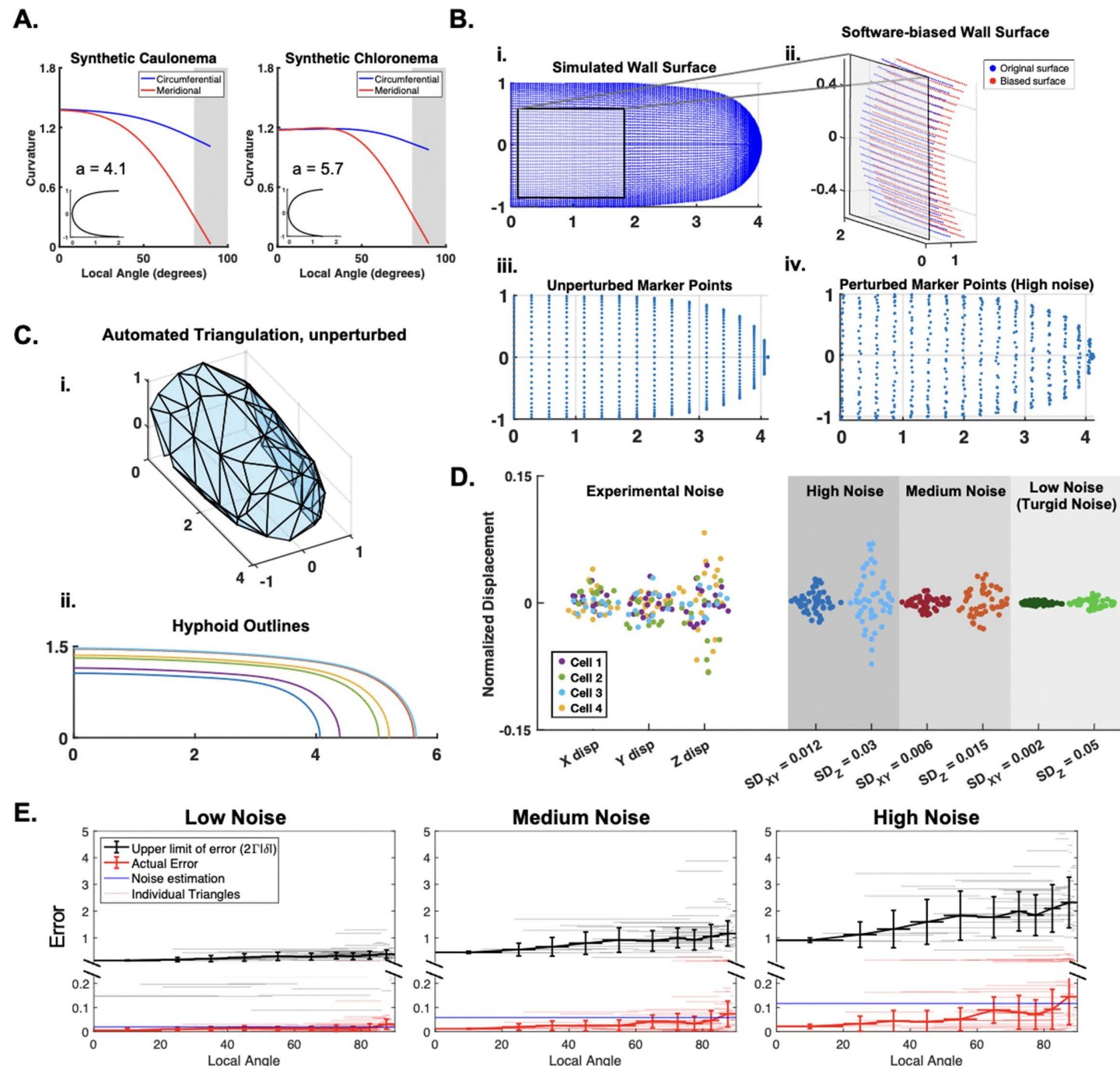

**Fig 3. Synthetic cell framework and noise estimation. A.** Synthetic caulonema and chloronema cell shape (insets) with circumferential and meridional curvature. The shape is obtained by the hyphoid-function fitting with a single parameter "a" ([Eq 11], S5 Fig). Gray region denotes the cell side. **B.** **(i)** Simulated cell surface **(ii)** Software-biased wall surface (red) compared against the original wall surface points (blue). Bias can occur outwardly or inwardly. **(iii).** Unperturbed cell marker points **(iv).** Perturbed cell marker points (high noise). **C. (i.)** Example synthetic cell triangulation with unperturbed marker points. **(ii).** Collection of synthetic caulonema cell outlines with different widths, to mimic size variation in reality. **D.** (Left) Noise estimation from experiments (see [Methods]). Data taken from n = 4 cells separated into X, Y, and Z displacement. Displacement was normalized by the cell radius. (Right) Replicated synthetic noise for three cases: high, medium, and low. Values are drawn from a normal distribution with mean = 1 and SD shown for each corresponding column. **E.** The relative error of $\lambda_1\lambda_2$ is shown in red. The theoretical upper limit of the error using the function $2\Gamma|\delta|$ is shown in black. For both values, the individual triangles and binned average are plotted (see legend). For each noise case, we estimated the upper limit of the displacement noise $|\delta|$ (blue line). For results on other triangulation sizes, see [S8 Fig].

setup and microscope capabilities. We have estimated the experimental setup noise by imaging the random movement of fluorescent beads on a cell (S1 Text, Sect 1.5). We found that the results followed a normal distribution, with the noise in the z-direction being twice that of the x-y-directions (Fig 3D, left). We set three noise levels on the unturgid cell using normal distributions with different standard deviations (Fig 3D, right). These can be added to the synthetic unturgid marker points to replicate noise on the marker points during the experiment (Fig 3B, iv). The largest noise case follows the standard deviation of the measured experimental noise. The smallest noise case is when there would only be noise from the software itself, so the unturgid and turgid marker point noises are equal. We estimated the software noise to have a standard deviation of 0.002 in the x-y-plane and standard deviation of 0.005 in the z-plane (Fig 3D, right). The noise values, denoted $\delta$, have all been rescaled by the cell radius: $\delta \rightarrow \bar{\delta}/\bar{L}$.

Using these $|\delta|$ values, we analyzed how error on a triangle compared to a triangle's theoretical upper limit of error [29]. We previously studied how triangle sensitivity is affected by its shape, leading to the relation: $\frac{\delta\|\mathbf{F}\|_2}{\|\mathbf{F}\|_2} \leq \Gamma|\delta|$ (see Sect 1.4 in S1 Text). To relate this upper limit to a measure of actual error, we used the fact that the eigenvalues of $\sqrt{\mathbf{F}^\mathsf{T}\mathbf{F}}$ are $\lambda_1$ and $\lambda_2$. Therefore, $\frac{\delta(\lambda_1\lambda_2)}{\lambda_1\lambda_2} = \frac{\delta\lambda_1}{\lambda_1} + \frac{\delta\lambda_2}{\lambda_2} \leq \frac{\delta\|\mathbf{F}\|_2}{\|\mathbf{F}\|_2} + \frac{\delta\|\mathbf{F}\|_2}{\|\mathbf{F}\|_2}$. This connects a measure of triangulation error with the derived analytical formula: $\frac{\delta(\lambda_1\lambda_2)}{\lambda_1\lambda_2} \leq \frac{2\delta\|\mathbf{F}\|_2}{\|\mathbf{F}\|_2} = 2\Gamma|\delta|$. Using the three noise cases just established, we determined that the theoretical upper limit of each triangle was always higher than the relative error of $\lambda_1\lambda_2$ (Figs 3E and S8). We will elaborate on this further in the Results.

## Results

Having established our surface morphology-based method to infer the spatial bulk modulus ($K_h$) (Figs 1, 2, Methods), we applied it to data from *Physcomitrium patens* tip-growing cells. However, there were still many parameters that needed to be taken into consideration. To test the inference method, we utilized experiments to inform simulation parameters such as the noise and cell morphology. Simulated cell morphology was matched to the two moss cell types, including the more tapered caulonema cell and the rounded chloronema cells (Fig 3A–3C). To organize the results, we primarily presented on the caulonema shape, with comparisons to the chloronema shape when different cell shapes affect the inference outcome. In addition, to test the consequence of different $K_h$ distributions, we consistently used cells with three cases of elastic moduli distributions: constant, nonlinearly, and linearly decreasing (see Methods). As a result of the study, we were able to determine the optimal triangulation size and cell sample size needed for the experiment. Finally, we returned to the experimental *P. patens* results with added verification by generating a theoretical reference map.

### Inference of elasticity gradient in single cells with no noise

**Sphere-fitting method is sensitive to cell shape:** We first assessed an inference method for the wall surface modulus ($Y$) published in [27] (see Methods for formulas). We tested this method by using the wall surface of synthetic caulonema and chloronema cells with varying elastic moduli distributions (constant, nonlinear, and linear S7 Fig). At the side, our estimate in both cell types were always within 5% error of the ground truth $Y_{side}$. However, at the tip of the cell, we found that $R_A$ and $R_B$ varied depending on the range of cell wall surface points used to fit the sphere (S7A and S7C Fig). While the more tapered caulonema cells with constant and linear elastic moduli had good inference of the modulus, the nonlinear graded cell inferences were all underestimated (S7B Fig). Strikingly, the chloronema cells gave non-physical results due to $R_A$ being larger than $R_B$, leading to negative values of $Y_{tip}$ (S7C and S7D Fig). Given the inconsistent results, we reasoned that the complexity of elastic deformation at the tip in rounder cells could not be directly captured by the change in tip radius. Although we acknowledge that this method could work for more tapered cells, we move forward with investigating the inference with our marker point method (Figs 1 and 2) to measure $K_h$.

**Marker point tracking can accurately infer elasticity with no noise:** Using our established method (Figs 1 and 2), the elastic stretch ratio, curvature, and tension parameters were all well inferred from the wall surface (S9 Fig). Consequently, the averaged error of $K_h$, denoted $\varepsilon_K$, was within 4% error in all three cases and in both caulonema and chloronema cells

(see S9A and S9B Fig, respectively). We also inferred $K_h$ using the triangle area change instead of $\lambda_s\lambda_\theta$, and the two results were virtually identical (S9 Fig, cyan result).

From here, we investigated how the size of the triangulation would affect the precision of our method with no noise. We present three triangulation cases with different averaged sizes, $\bar{\Delta}_A/\bar{L}^2$, where $\bar{\Delta}_A$ is the mean triangle area, and $\bar{L}$ is the cell radius. The three cases are visually distinguishable, with different distributions of normalized triangle areas (S10A Fig). We found that the error in $K_h$ in the large triangulated cell came from the increased deviations from the ground truths at the tip as the triangulation size increases (S10B–D Fig). To put this back into the perspective of the cell wall surface, we mapped the trend of $\lambda_s\lambda_\theta$ on each triangulation (Fig 4A). We can observe the smaller $\bar{\Delta}_A/\bar{L}^2$ case gives the best resolution of the gradient of $\lambda_s\lambda_\theta$.

### Inference of elasticity gradient in single cells with noise

To quantify levels of noise, primarily from the marker point localization, we used our experimental setup to obtain an estimate (Fig 3D, Methods). Our reported standard deviation values representing the noise $\delta$ are normalized to our cell radius $\bar{L}$, namely $\delta/\bar{L}$ (Fig 3D). After adding them to our synthetic cell triangulations, we first observed the effect of noise on the single cell triangulation level (Fig 4B–4D). At the lowest noise level (Fig 3D), we can see the loss of the smooth gradient of $\lambda_s\lambda_\theta$ and an overall increase in the averaged error, $\varepsilon_K$, compared to no noise. Notably, the correlation between error and triangulation size was disrupted at the low noise level, as the smallest triangulation exhibited the greatest error (Fig 4B). We can observe that the small triangulated cell surface becomes more heterogeneous in comparison to the other two sizes. This patterning becomes more pronounced in all cases at the next two levels of noise (Fig 4C–4D). There is no clear trend of $\varepsilon_K$ among different noise levels due to the randomness of noise. However, the large triangulation is the most stable across all noise levels. This agrees with our theoretical analysis using the upper error bounds computed on each triangle (S8 Fig and Sect 1.4 in S1 Text). The smaller triangles are more sensitive to noise (S8C Fig). The logic follows in practice, because as the triangulation gets smaller, the ratio of marker point displacement from noise to triangle size is smaller. Considering that the results of a single cell can suffer due to the randomness of noise, we next explored the use of multiple cells to infer the mean-property against the ground truth.

**Effect of multiple cell data on inference accuracy:** Assuming a mean-distribution of elastic moduli exists among a group of cells, we analyzed how the inference from multiple cells helps to approach the underlying property. To implement this, we leveraged our synthetic cell data to create multiple hyphoid cells with the same ground truth elastic moduli distributions (Fig 3C, ii). Each cell was given a random width within a range to replicate different experimental cell sizes. Then we applied our inference procedure to each cell with different triangulations and random noise, and averaged the $K_h$ results at increasing cell sample sizes (see Sect 1.7 in S1 Text).

We first analyzed the trend when the lowest amount of noise was added (S11 Fig). As anticipated, we saw that the mean relative error had a sloping behavior with an initial drop as the trial number went up. As an optimization problem of error versus trial number, we reported the "knee" of the curve and corresponding averaged error at that point ($\bar{\varepsilon}_K$) [43]. In the lowest noise level, we observed the disrupted triangulation trend seen in Fig 4, but with overall improved inference (S11 Fig). Although, the medium triangulation performed the best overall at the low level of noise, once we increased the noise to the medium level, the large triangulation started performing the best (S12 Fig). The overall error was significantly increased at the highest level of noise, but the largest triangulation remained with the best result (S13 Fig). We further found that we could effectively increase our data without adding additional samples by adding another triangulation on the same cell (S1 Text, Sect 1.7). Although the decrease in the overall error was not significant, we noticed that this helped decrease the knee value, the optimal point of sample size versus error (S13 and S14 Figs). We applied this two triangulation method to our subsequent analyses. Ultimately, we concluded that $n = 10$ cells, each with two triangulations, was an optimal sample number given the level of noise from experiments, for minimizing both the error and sample size.

PLOS Computational Biology

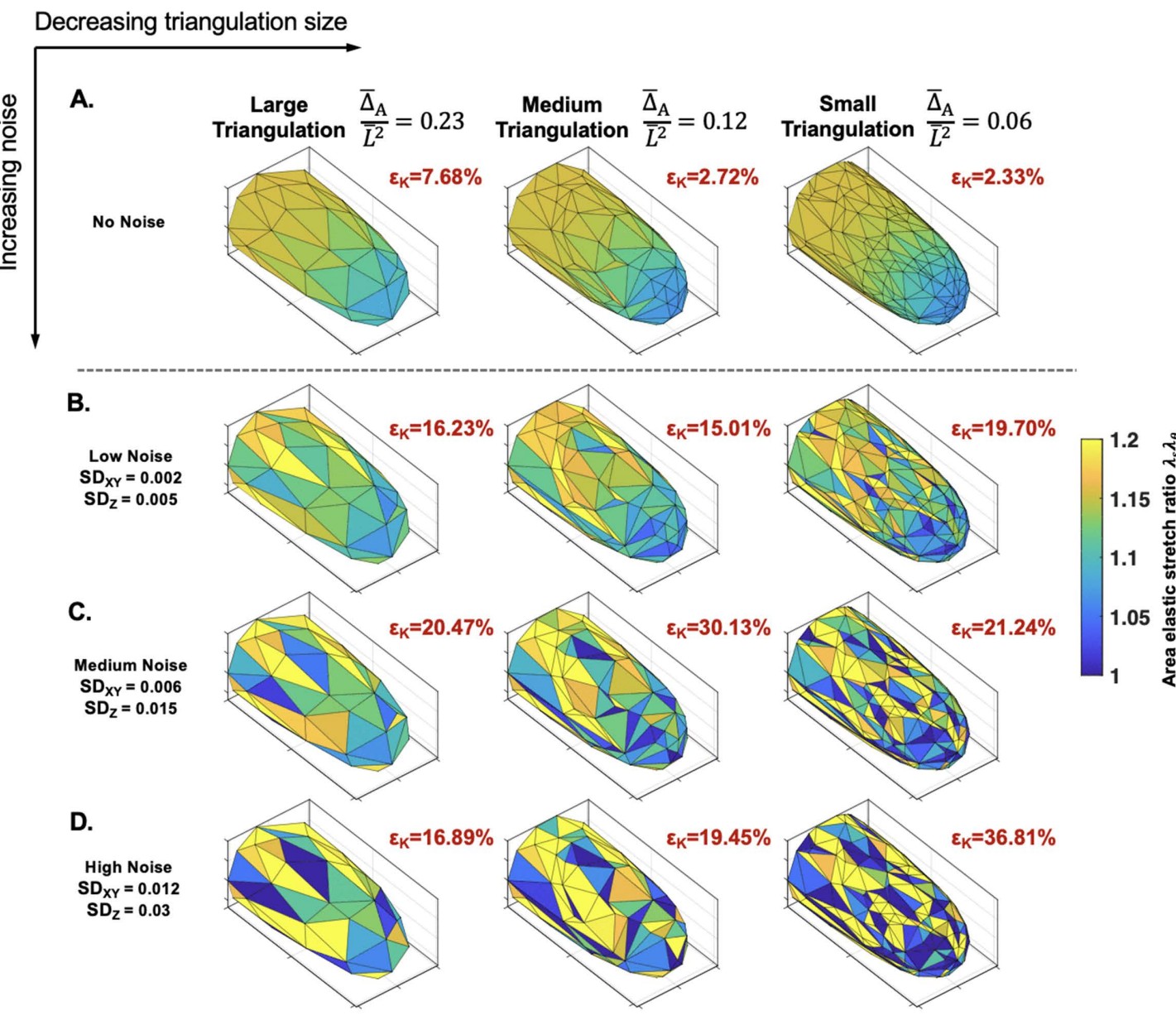

**Fig 4. Effects of triangulation size and noise on a single cell: For each triangulation size, the same triangulated cell is shown with increasing noise on the marker points: A. no noise, B. low noise, C. medium noise, and D. high noise.** Each triangle is colored by their area elastic stretch ratios: $\lambda_\theta\lambda_s$. Triangles with values outside the bounds of the color bar are assigned the colors at the boundary. The $\varepsilon_K$ value shows the corresponding $K_h$ error for each cell, and are calculated after filtering triangles with $\lambda_s\lambda_\theta < 1$. All cells shown have a ground truth of $K_h = 5$, constant.

**Sensitivity of gradient recovery against large perturbations:** The importance of our method is its ability to capture a spatial distribution of elastic modulus. Through averaging the standard $n = 10$ cells, the nonlinear gradient was recovered in all three triangulation cases and re-established on the synthetic cell wall surface (Fig 5A, $K_h$). The corresponding averaged error distribution (Fig 5A, $\bar{\varepsilon}_K$) indicated that the large and medium triangulations typically suffered at the sharp transition region. Interestingly, the small triangulation has higher error at the side and tip, but has better accuracy at the

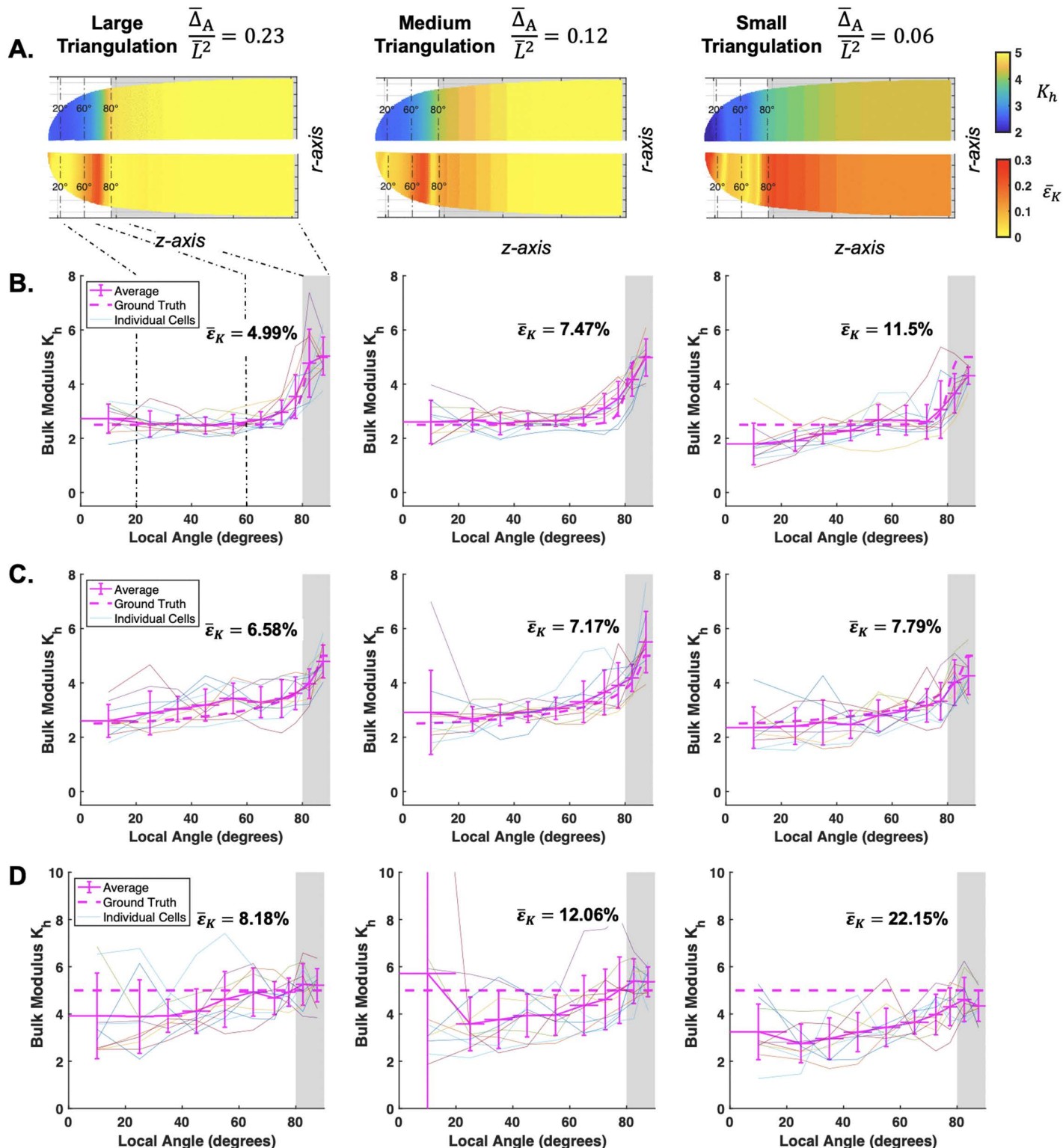

**Fig 5. Inference of bulk modulus with high noise can be recovered with multiple cell averaging. A.** Averaged result of n = 10 cells interpolated back onto a synthetic caulonema cell (top). Averaged error ($\bar{\varepsilon}_K$) shown on the synthetic cell (bottom). Dashed lines denote local angle reference markers on the cell. The spatial results along the local angle are shown for a **B.** nonlinear, **C.** linear, and **D.** constant case. Each individual cell result and ground truth bulk modulus are plotted behind. Modulus values are non-dimensionalized ($K_h = K_h/\bar{P}\bar{L}$).

transition area, improving its error overall (see also S14B and S14C Fig insets). The linear results performed similarly well to the nonlinear, with overall better inference since the methodology does not suffer due to a sharp transition (Fig 5B and 5C).

In cells with constant elastic moduli, we observed a false gradient result due to under-estimation of $K_h$ at the tip (Fig 5D). This trend, starting in the small triangulated cells, was amplified across all test cases as the noise increased (see S12A and S13A Figs insets). Given the frequency of these false results, the ground truth constant distribution was not fully recovered even after obtaining the average inference of 10 cells (Fig 5D). We reasoned that the overall under-estimation at the tip was caused by $\lambda_s\lambda_\theta$ approaching 1 there, making $K_h$ more susceptible to perturbations (see Eq 8). The constant case of $K_h = 5$ has lower values of $\lambda_s\lambda_\theta$ at the tip, compared to the tip of the graded cases (see S9A Fig, first column). When noise is added, the elastic stretch ratios fall above or below the ground truth. But at a ground truth of $\lambda_s\lambda_\theta \approx 1.06$, many values will fall below 1, causing negative $K_h$ values. After filtering the triangles whose $\lambda_s\lambda_\theta < 1$, we are left with triangles that fall above the ground truth. This unbalance of higher elastic stretch will correlate with an underestimation of $K_h$. In general, there is also poorer inference at the tip due to there being inherently less triangles available there compared to the cell side. Generally, the coupling effect between $K_h$ and the cell geometry means that more tapered cells will generally suffer more since they are under less tension and thus have lower values of elastic stretch. We confirmed this in the rounder chloronema cells, showing that although we still observed under-estimation of $K_h$ at the tip, the overall gradient is flatter (S15A Fig).

### Simulated global error map verifies the spatial *P. patens* cell wall elasticity profile

Our parameter sensitivity study informed us that 10 cells was sufficient in recovering a gradient. Therefore, we collected 10 samples each of *P. patens* caulonema and chloronema cells. Although we were limited in our marker point coverage experimentally, we did aim to adopt a larger triangulation given the results from our study. The results of our experimental triangulations showed that we were within a medium to large triangulation size (Fig 6A). After averaging the bulk modulus result across all 10 cells, we observed a spatial change within two folds in both cell types (Fig 6B). Similar to Fig 5A, we reported the results back on their respective simulated shapes (Fig 6C). The corresponding averaged curvature, rescaled tension, and elastic stretch ratio results also show reliable convergence of the circumferential and meridional properties at the tip (Fig 6D). We noted that $\kappa_s$ in the last bin (gray region) did not reach 0 due to averaging within the bin. Therefore, the ratio of circumferential tension to meridional tension in the cell side region is below 2.

Our observation in the previous section showed a false gradient inference from a cell with a constant $K_h = 5$. Therefore, we wanted to verify that the result from *P. patens* was not false by using simulated cells with the same ground truth $K_h$ gradient. To reach a more general result applicable to a broader set of experimental set-ups, we also investigated a range of $K_h$ constant and gradient cases.

**Endpoint ratio mapping across different triangulation sizes and noise levels:** We created a set of simulated cells with unique pairs of surface $K_h$ side and tip values, including constant and gradient cases, which were given a nonlinear distribution (S16 Fig). For each case, we inferred the standard 10 cells repeatedly, and used their averaged inference to calculate the inferred endpoint ratio: $K_{side}/K_{tip}$ (S1 Text, Sect 1.7). The final results were interpolated across the grid (Fig 7). At the lowest level of noise, the three triangulation resolution results are similar (Fig 7A). The top right corner of each map has the lowest values of elastic stretch, and we can see the contour distortions start there in the small triangulation map as noise is increased (Fig 7B). This begins to affect the large and medium triangulations when the noise is further increased (Fig 7C). We further confirmed that the chloronema results with the same range of $K_h$ performed better overall (S17 Fig). This is again due to the coupling between tension and $K_h$, causing higher elastic stretch ratios and thus, more accurate results (S16 Fig).

To interpret the mapping in more detail, we can see how the problem of a false gradient corresponds with the contour expansions. For example, notice the invasion of the contour group $K_{side}/K_{tip} = 1.4$ into the upper part of the 1.2 contour

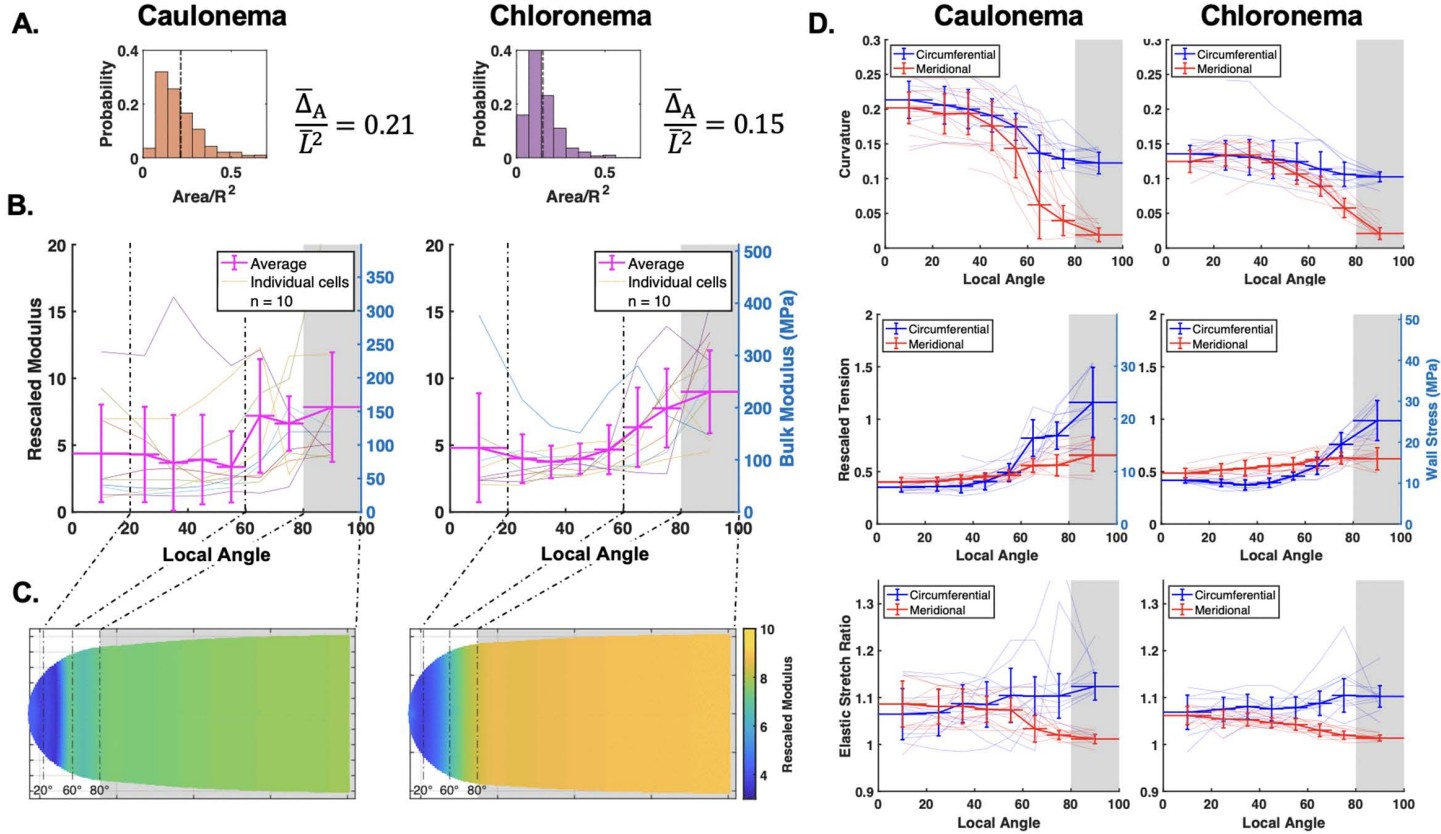

**Fig 6. Inference of *P. patens* caulonema and chloronema wall surface parameters.** Each part displays the caulonema result on the left and chloronema result on the right. **A.** Experimental triangulation sizes shown in histogram. Values representing the mean rescaled size, $\overline{\Delta}_A/\overline{L}^2$, are shown. **B.** Spatial average bulk modulus result for n = 10 cells. Individual cell results results are plotted underneath. Left axis shows the rescaled values while the right axis shows the values in MPa. **C.** Results of the rescaled modulus are interpolated onto the respective synthetic caulonema and chloronema cells. Dashed lines denote local angle reference markers. **D.** Corresponding average curvature, tension, and elastic stretch ratio results. For the tension result, the right axis shows the wall stress value in MPa.

group in Fig 7C, large triangulation. The ground truth value on the diagonal is $K_{side}/K_{tip}$ = 1, and the mapping shows that for a $K_{side}$ >6, the inference result reports 1.4. With that same logic, the confidence range is reduced in the medium triangulation, and further reduced in the small triangulation (Fig 7C). In the rounder chloronema cells, the large triangulation can infer a constant $K_h$ up until the uppermost corner (S17C, left).

**Two error quantification mappings reveal regions of reliable inference:** To quantify a direct value of reliability, we overlaid the values of the averaged error ($\bar{\varepsilon}_K$) of the inference and the endpoint ratio error, $\bar{\varepsilon}_{K_{side}/K_{tip}}$ (Fig 8) The center of the averaged error map for caulonema and chloronema cells is consistently within 10% error, reinforcing our results from Fig 5 (Fig 8A, 8B). But as suspected, the upper right corner of the phase map where the moduli are high and the elastic stretch ratios are low, has higher error values in the small triangulation map. On the other hand, the endpoint ratio error more clearly reflects the under-estimation of $K_{side}$ and/or $K_{tip}$ (Fig 8C, 8D). The absolute value is shown, but there is a transition from positive ratio error to negative along the middle of the map. As the ratio increases towards the bottom right of each map, we found that the method may not accurately infer the nonlinear distribution's sharp transition due to the bin size. This typically caused under-estimation of $K_{side}$, leading to negative $\bar{\varepsilon}_{K_{side}/K_{tip}}$. Error in the top right came from underestimation of $K_{tip}$ (Fig 8C, 8D). The edge of the lowest

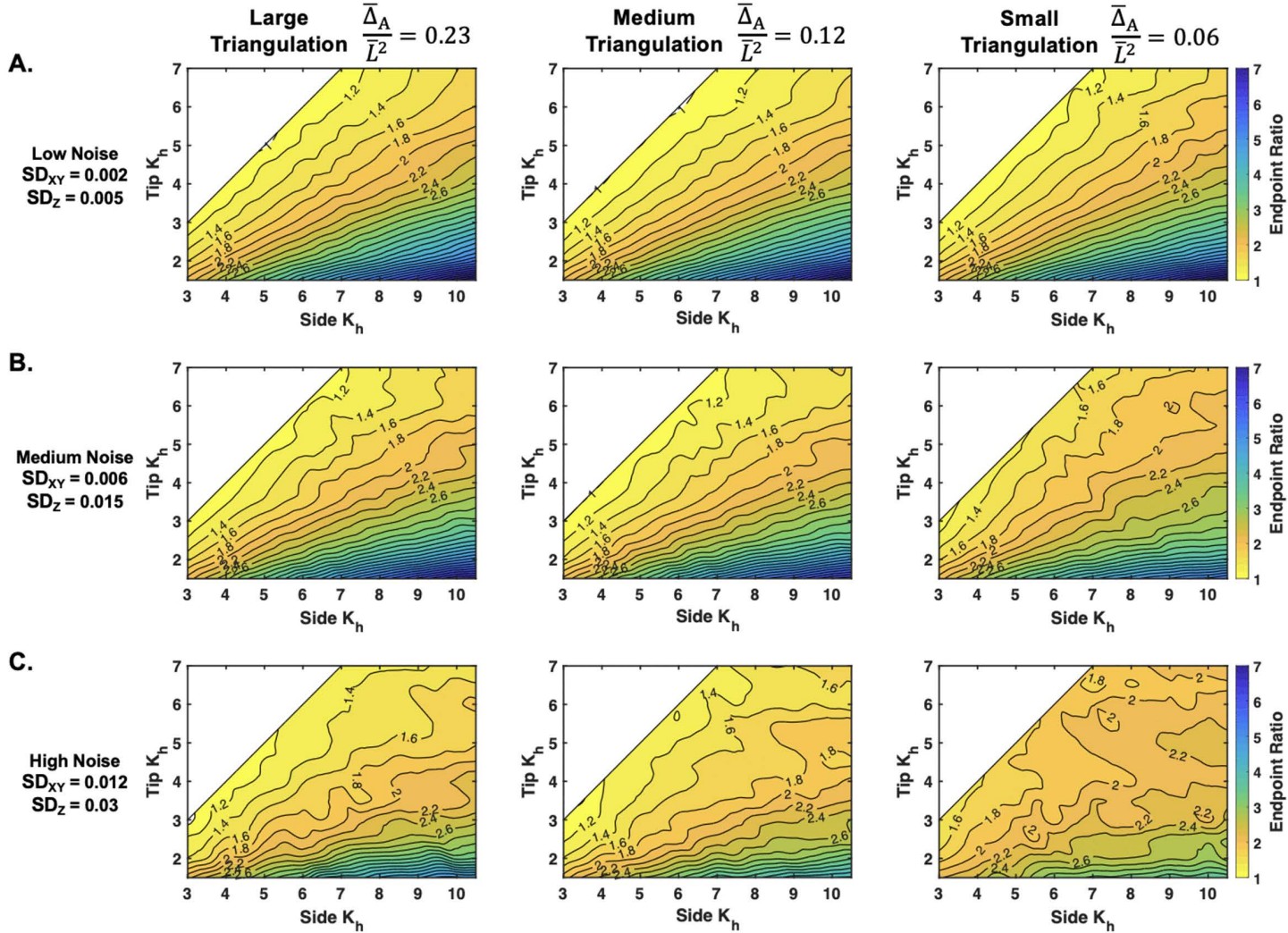

**Fig 7. Inferred endpoint ratio results across different triangulation sizes and noise levels from a synthetic caulonema cell.** Contour map depicts the inferred endpoint ratio levels ($K_{side}/K_{tip}$) across a range of $K_{side}$ and $K_{tip}$ values at the three triangulation sizes and different noise levels: **A.** low noise, **B.** medium noise, and **C.** high noise. Results come from repeating the averaging of the standardized 10 cells to remove random effects (see S1 Text). Modulus values are non-dimensionalized ($\bar{K}_h = \bar{K}_h/\overline{PL}$).

level set ($K_{side}/K_{tip}$ = 1.2) can be seen corresponding to about a 20% endpoint ratio error (Fig 8C, black 1.2 line and white dashed lines). We translated this to be a reasonable value of error bordering the false gradient problem, and highlighted this boundary in both error maps (Fig 8, white dashed lines). We also used 5% error as a stricter experimentally acceptable error boundary, which showed the triangulation size trend through the increase of the bounded area from small to large triangulation (Fig 8C, 8D white dotted lines). As expected, at lower levels of noise, the thresholds of 5% and 20% error are expanded (S18 Fig and S19 Fig).

Assuming the highest amount of noise, we took the endpoint ratio results from *P. patens* caulonema and chloronema cells (Fig 6B) and placed them on the generated error maps (yellow stars, Fig 8). Since our experimental results used triangulation sizes between medium and large (Fig 6A), we directed our results onto those respective maps. For both cell types, the experimental results fell within 20% error for both the averaged error and the endpoint ratio error. Additional

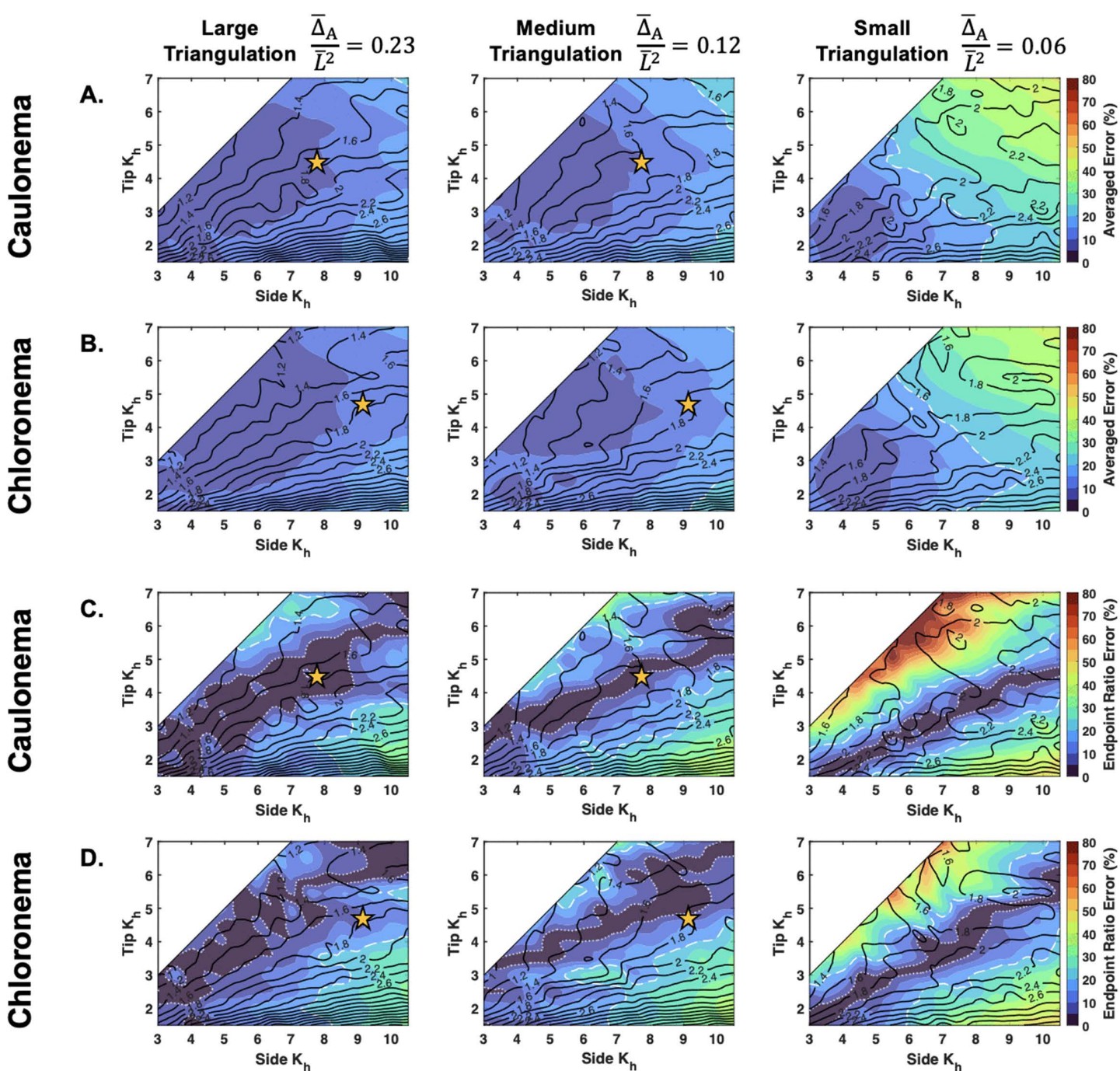

**Fig 8. Two error quantifications for caulonema and chloronema cell bulk modulus inference with high noise.** Averaged error ($\bar{\varepsilon}_K$) for **A.** caulonema and **B.** chloronema cells. Relative absolute value endpoint ratio error ($\bar{\varepsilon}_{K_{side}/K_{tip}}$) for **C.** caulonema and **D.** chloronema cells. In all parts, the underlying contours from Fig 7 (caulonema) and S17 Fig (chloronema) are shown. The white dashed line denotes the boundary of 20% error and the white dotted line denotes the boundary of 5% error. Results come from repeating the averaging of the standardized 10 cells to remove random effects (see S1 Text). Yellow stars indicate the $K_{side}/K_{tip}$ location of the caulonema and chloronema experimental results shown in Fig 6B, within the range of triangulation sizes used. Modulus values are non-dimensionalized ($K_h = \bar{K}_h/\bar{P}\bar{L}$).

verification that our result is not a false gradient can be seen by the contours. Our results are in a contour that does not invade a contour along the diagonal.

## Discussion

The spatial distribution of cell wall elasticity in tip-growing cells plays a key role in determining how plant cell walls deform. Quantifying this elasticity will be an important input into tip growth models. Here, we presented a systematic analysis of a new method to measure the elastic properties of the cell wall. By using a surface triangulation method to connect marker points across the cell surface, we were able to adapt the axisymmetric mechanical model to measure the wall elastic moduli (Figs 1 and 2). Through analytical and numerical parameter studies, we found that a coarser triangulation size was more robust against noise and that a sample size of 10 cells was necessary to recover the ground truth in all cases with noise. Using that number, we inferred the spatial bulk modulus in *P. patens* caulonema and chloronema cells (Fig 6). To verify this result, we created a reference map encompassing a range of elastic moduli distributions. Throughout our study, we presented dimensionless variables with the aim that this method can be adapted to any walled system. This could open many avenues for experimentalists to incorporate measured cell wall properties into tip growth mechanical modeling.

Modeling work has postulated that sharp gradients of wall mechanics occur in a small region between the cell side and tip, which we call the transition region [18–20]. Adapting techniques to confirm this in an experimental framework has been challenging because it requires precise tracking of the cell wall deformation along the meridian. The primary significance of our method is that it aims to measure the continuous wall properties along marker point positions in this transition region. To justify the use of our method experimentally, we performed a parameter sensitivity study on simulated cells with ground truth elastic moduli distributions (Fig 3). Since we did not know how cell morphology may affect the inference of these cell wall properties, we utilized the *P. patens* cells at our disposal. The canonical caulonema and chloronema cell types provided a point of comparison between a tapered and round morphology, respectively (Fig 3). Using our marker point methodology, we were able to verify good inference accuracy in both synthetic cell types with no noise (S9 Fig). Importantly, we found that the method was able to accurately measure the cell wall elasticity changes in the transition region.

Another advantage of using simulated cells was to determine the optimal triangulation size and sample size given different levels of noise. With dimensionless parameters, the presented results can be interpreted to any walled system with different osmotic pressures, cell sizes, and experimental marker point techniques. These dimensionless parameters include the noise on the marker points $\delta/\bar{L}$, triangulation resolution $\bar{\Delta}_A/\bar{L}^2$, and the dimensionless elastic moduli $K_h = \bar{K}_h/P\bar{L}$. $\delta/\bar{L}$ was estimated from our experimental setup (Fig 3D), and will depend on the cell width. For that reason, as well as potential experiment setup differences, we presented different levels of noise throughout (Figs 4, 7, and 8). The differences in experimental setup, will also be an experimental constraint on the triangulation size. We used fluorescent microspheres with application through liquid flow onto the tip-growing cell (Fig 1A), although we have seen other dependable techniques [20,23]. Given the uncertainties of the marker point density, we studied the effect of different triangulation sizes on the inference. Our previous findings showed that a coarser spacing of marker points was more stable against noise [29], which we first verified theoretically (Figs 3E and S8). Our analytical results further concluded that using a larger triangulation gave the most stable results (Figs 4–8). Thus, our recommendation would be to use a larger triangulation, relative to the cell size. The final triangulation sizes can be benchmarked by the dimensionless parameter representing the triangle areas, $\bar{\Delta}_A/\bar{L}^2$ (Figs 4–8).

We did not infer the distribution of $\mu_h$ experimentally since the inference equation (Eq 7) tells us that there will be instabilities for any cell with isotropic properties at the tip. As a result, the Poisson's ratio cannot be inferred yet by the current method. In all synthetic cells, we adopt a previous assumption that the Poisson's ratio is constant along the cell wall for simplicity [20,45]. On the other hand, $K_h$ is inferable at the tip but also suffers from an instability when the area stretch ratio $\lambda_s\lambda_\theta \sim 1$ (Eq 8). As shown in the equation, a close-to-zero area strain $\lambda_s\lambda_\theta - 1$ can lead to large perturbations on $K_h$,

causing inaccurate inference or even non-physical negative values of $K_h$. While we have applied a universal filtering process to rule out triangles with negative $K_h$ values, those at the tip regions are more susceptible to this instability due to the lower tension that often leads to smaller area strain there.

In cells with a constant distribution, we see that the combined instability and filtering process leads to an underestimation of $K_h$ at the tip. This caused the inference of a false gradient. To encapsulate the occurrence of a false gradient, we mapped the problem using generated contour and error maps (Figs 7 and 8). Using the maps, we confirmed that most $K_h$ distributions, including our *P. patens* results (Fig 6), could be recovered using multiple cell samples and a larger $\bar{\Delta}_A/\bar{L}^2$. Although we acknowledge the reference maps may not be applicable to morphologies at the more extreme ends of the shape spectrum [4,37], the generalized interpretation can be made with a tapered and round tip morphology. In general, we also found that since rounder cells are under more tension, they will exhibit higher values of elastic stretch and therefore be better inferred. Previously published reports of cell wall deformation demonstrate that most walled cells have sufficient elastic stretch [9,23,27,29]. This supports the feasibility of this method to infer the spatial elasticity in other walled organisms.

To compare the mechanical parameters with previously published results, we estimated the units of the *P. patens* bulk modulus. We approximated the turgor pressure in *P. patens* to be $\bar{P} \approx 0.75$ MPa, similar to findings in fungal tips and brown algae [9,27]. Those same publications also provide estimates for the cell wall thickness similar to measurements made in *P. patens* at $h \approx 260$ nm [44]. We first estimated the wall stresses, which are the wall tensions divided by the wall thickness, $\bar{\bar{\sigma}}_{s,\theta} = \frac{\bar{P}\cdot\bar{L}}{h}\sigma_{s,\theta}$. Using an average of $\bar{L}_{caul} \approx 7\ \mu m$ and $\bar{L}_{chlo} \approx 9\ \mu m$, our estimates of wall stress were consistent with previous reports (Fig 6D, see right axis of middle pane) [4,9,20,45]. Likewise, we estimated the bulk modulus using the equation $\bar{K} = \frac{\bar{P}\bar{L}}{h}K_h$, and reported this on the right axis of Fig 6B. Seeing that the gradient estimations are comparable to fungal cell walls [27], we are more inclined to understand the significance of the wall elastic moduli profile among different cell types. We will also explore the method limitations shown here, specifically the measurement of the shear modulus. Knowing both elastic moduli will be essential to fully describe the wall material and model how the mechanics of the wall modulate tip cell growth and morphogenesis.

## Supporting information

**S1 Text.** Supplemental text includes Sect 1.1: Experimental protocols and moss imaging, Sect 1.2: Wall surface data processing, and marker point matching and projection, Sect 1.3: Computation of curvatures from the wall surface, Sect 1.4: Triangle error analysis, Sect 1.5: Experiment noise estimation, Sect 1.6: Range and distribution of elastic moduli inputs to generate synthetic cells, and Sect 1.7: Data integration and multiple cell averaging.
(PDF)

**S1 Fig. Conversion of experimental data into MATLAB and inference method details. A.** Example of marker point localization in ImageJ. Sub-pixel resolution locations of fluorescent points are identified by the RS-FISH plugin. **B.** Example of cell wall outline localization using the Ridge Detection plugin in ImageJ. **C.** Demonstration of marker point matching between the two configurations in MATLAB. Matching is done manually through visual confirmation of the two sets of marker points. Additional global displacement of one set of marker points may be needed to aid in the visual matching (bottom). **D.** Identification of tip point is done using the wall outline data and the max x-value. **E.** Demonstration of marker point projection from two views (top). Histogram depicting distances between the bead and wall outline for both the turgid and unturgid wall outlines (bottom).
(TIF)

**S2 Fig. Comparison of triangulation techniques. A.** Surface triangulation using 3D Delaunay algorithm. **B.** Stereographic-projection method presented in our main text, on the same set of marker points. For both **A.** and **B.**, the triangle color indicates the value of its compactness score: $4\pi$Area/Perimeter$^2$. The triangles outlined in red are triangles filtered out using our

sensitivity analysis algorithm. **C.** Histogram of compactness score for the two triangulations. The histogram outline in red is the result after filtering out the triangles shown in red in **A.** and **B.** respectively.
(TIF)

**S3 Fig. Defining local angle on the wall surface points and triangulation. A.** Local angle of each cell wall outline point calculated using $\alpha = \cos^{-1}(\hat{n} \cdot \hat{z})$. **B.** Triangulation map of local angle. **C.** (Top) Example triangle with local angle range. (Bottom) Example mapping of data along the local angle. Each triangle is represented by a line showing its angle range, and then it is sorted into its bins (example bin in gray).
(TIF)

**S4 Fig. Confirmation of chosen spacing size in curvature calculation. A.** Mean curvature results at 5 different spacing levels shown. **B.** The corresponding probabilities of mean curvature at the 5 spacing levels. We have chosen the bolded yellow line, spacing = 3 pixels.
(TIF)

**S5 Fig. Experimental caulonema and chloronema hyphoid parameter fitting. A.** Caulonema fitting, best fit at $a = 4.1$. **B.** Chloronema fitting, best fit at $a = 5.7$. (Top) Score of fit versus the parameter $a$ shown in the equation below. (Bottom) Best fit hyphoid curve shown in red on top of 6 experimental cell outlines (both sides of each cell outline are included).
(TIF)

**S6 Fig. Effect of different $\mu_h$ gradients in relation to the $K_h$ gradient.** Three gradient cases are studied: the ratio of the side to the tip are equal for both $K_h$ and, is constant, and the relative ratio of $K_h$ and is equal. **A.** The turgid cell shapes of the three cases are compared. Inset shows the $K_h$ gradient. **B.** The respective gradients shown. **C.** The respective elastic stretch ratios for the three cases shown.
(TIF)

**S7 Fig. Sphere-fitting method to measure synthetic caulonema and chloronema tip surface modulus. A.** Sphere radius of the turgid (blue) and unturgid sphere fitting (red) versus the maximum degrees of the synthetic cell outline used to fit the sphere. The three gradient cases are shown for each cell type. Visual sphere fit result examples are shown in the inset. **B.** Corresponding modulus value calculated by $Y_{tip} = \frac{R_B}{2(R_B - R_A)/R_A}$ where $R_A$ is the unturgid sphere radius and $R_B$ is the turgid sphere radius. Insets on some graphs show re-focused view. Chloronema results are shown in **C.** and **D.** Modulus values are non-dimensionalized.
(TIF)

**S8 Fig. Principal stretch ratio error is below theoretical limit and comparable to displacement noise.** For the three triangulation cases **A.** large, **B.** medium, and **C.** small, the relative error of $\lambda_1 \lambda_2$ is shown in red. The theoretical upper limit of the error using the function $2\Gamma|\delta|$ is shown in black. For both values, the individual triangles and binned average are plotted (see legend). For each noise case, we estimated the upper limit of the displacement noise $|\delta|$ (blue line).
(TIF)

**S9 Fig. No noise inference of geometric parameters in synthetic caulonema and chloronema cells.** Inference results of a single **A.** caulonema and **B.** chloronema cell with no noise. Each part shows an inference on cell's with a constant, nonlinear, and linear distributions. In each case, the elastic stretch ratio, curvature, and tension inferences are shown with their corresponding ground truth values. The rightmost panels depict the bulk modulus inference along with their relative error from the ground truth, $\varepsilon_K$. The cyan curve depicts the inference of $K_h$ calculated with $S_B/S_A$. Note that in the linear case, the curve will be linear when plotted as a function of $z$, instead of $\alpha$. Values are non-dimensionalized ($\kappa_{s,\theta} = \bar{\kappa}_{s,\theta}\bar{L}$, $\sigma_{s,\theta} = \bar{\sigma}_{s,\theta}/\bar{P}\bar{L}$, $K_h = \bar{K}_h/\bar{P}\bar{L}$).
(TIF)

**S10 Fig. Triangulation size analysis with no noise. A.** Representative triangulation on a synthetic caulonema cell and histogram of normalized triangle areas. Dashed line on histogram represents mean area displayed above to define each triangulation size: $\bar{\Delta}_A/\bar{L}^2$. **B.** Corresponding elastic stretch results from the three different triangulations. **C.** Corresponding tension results from the three different triangulations. **D.** Corresponding bulk modulus results from the three different triangulations. The average relative error ($\varepsilon_K$) of the inference is shown. Values are non-dimensionalized ($\sigma_{s,\theta} = \bar{\sigma}_{s,\theta}/\bar{P}\bar{L}$, $K_h = \bar{K}_h/\bar{P}\bar{L}$).
(TIF)

**S11 Fig. Low noise caulonema trial analysis.** Average relative error of the bulk modulus inference for **A.** constant, **B.** nonlinear, and **C.** linear gradients versus the number of trials for three triangulation cases. In each case, $n$ trials are chosen from 100 total cells, and this is repeated 100 times to obtain the standard deviation and mean shown in each graph. The corresponding 100 cell rescaled bulk modulus inferences are shown in each inset on top of the ground truth bulk modulus in magenta. Additionally, the knee of the curve is plotted along the mean, with its averaged error ($\bar{\varepsilon}_K$) at that trial number.
(TIF)

**S12 Fig. Medium noise caulonema trial analysis.** Average relative error of the bulk modulus inference for **A.** constant, **B.** nonlinear, and **C.** linear gradients versus the number of trials for three triangulation cases. In each case, $n$ trials are chosen from 100 total cells, and this is repeated 100 times to obtain the standard deviation and mean shown in each graph. The corresponding 100 cell rescaled bulk modulus inferences are shown in each inset on top of the ground truth bulk modulus in magenta. Additionally, the knee of the curve is plotted along the mean, with its averaged error ($\bar{\varepsilon}_K$) at that trial number.
(TIF)

**S13 Fig. High noise caulonema trial analysis.** Average relative error of the bulk modulus inference for **A.** constant, **B.** nonlinear, and **C.** linear gradients versus the number of trials for three triangulation cases. In each case, $n$ trials are chosen from 100 total cells, and this is repeated 100 times to obtain the standard deviation and mean shown in each graph. The corresponding 100 cell rescaled bulk modulus inferences are shown in each inset on top of the ground truth bulk modulus in magenta. Additionally, the knee of the curve is plotted along the mean, with its averaged error ($\bar{\varepsilon}_K$) at that trial number.
(TIF)

**S14 Fig. High noise (two triangulation) caulonema trial analysis.** Average relative error of the bulk modulus inference for **A.** constant, **B.** nonlinear, and **C.** linear gradients versus the number of trials for three triangulation cases. In each case, $n$ trials are chosen from 100 total cells (with two triangulations each S1 Text), and this is repeated 100 times to obtain the standard deviation and mean shown in each graph. The corresponding 100 cell rescaled bulk modulus inferences are shown in each inset on top of the ground truth bulk modulus in magenta. Additionally, the knee of the curve is plotted along the mean, with its averaged error ($\bar{\varepsilon}_K$) at that trial number.
(TIF)

**S15 Fig. High noise (two triangulation) chloronema trial analysis.** Average relative error of the bulk modulus inference for **A.** constant, **B.** nonlinear, and **C.** linear gradients versus the number of trials for three triangulation cases. In each case, $n$ trials are chosen from 100 total cells (with two triangulations each, see S1 Text), and this is repeated 100 times to obtain the standard deviation and mean shown in each graph. The corresponding 100 cell rescaled bulk modulus inferences are shown in each inset on top of the ground truth bulk modulus in magenta. Additionally, the knee of the curve is plotted along the mean, with its averaged error ($\bar{\varepsilon}_K$) at that trial number.
(TIF)

**S16 Fig. Subset of caulonema and chloronema cell outlines and elastic stretch ratio.** For each subplot, the turgid cell outlines of the respective caulonema and chloronema cells are plotted (left axis). The circumferential (blue), meridional (red), and area stretch ratio (black) are plotted on top (right axis).
(TIF)

**S17 Fig. Inferred endpoint ratio results across different triangulation sizes and noise levels from a synthetic chloronema cell.** Contour map depicts the inferred endpoint ratio levels ($K_{side}/K_{tip}$) across a range of $K_{side}$ and $K_{tip}$ values at the three triangulation sizes and different noise levels: **A.** low noise, **B.** medium noise, and **C.** high noise. Results come from repeating the averaging of the standardized 10 cells to remove random effects (see S1 Text). Modulus values are non-dimensionalized ($K_h = \bar{K}_h/\bar{P}\bar{L}$).
(TIF)

**S18 Fig. Two error quantifications for caulonema and chloronema cell bulk modulus inference with low noise.** Averaged error () for **A.** caulonema and **B.** chloronema cells. Relative absolute value endpoint ratio error $\bar{\bar{\varepsilon}}_{K_{side}/K_{tip}}$) for **C.** caulonema and **D.** chloronema cells. In all parts, the underlying contours from Fig 7 (caulonema) and S17 Fig (chloronema) are shown. The white dashed line denotes the boundary of 20% error and the white dotted line denotes the boundary of 5% error. Results come from repeating the averaging of the standardized 10 cells to remove random effects (see S1 Text). Modulus values are non-dimensionalized ($K_h = \bar{K}_h/\bar{P}\bar{L}$).
(TIF)

**S19 Fig. Two error quantifications for caulonema and chloronema cell bulk modulus inference with medium noise.** Averaged error () for **A.** caulonema and **B.** chloronema cells. Relative absolute value endpoint ratio error ($\bar{\bar{\varepsilon}}_{K_{side}/K_{tip}}$) for **C.** caulonema and **D.** chloronema cells. In all parts, the underlying contours from Fig 7 (caulonema) and S17 Fig (chloronema) are shown. The white dashed line denotes the boundary of 20% error and the white dotted line denotes the boundary of 5% error. Results come from repeating the averaging of the standardized 10 cells to remove random effects (see S1 Text). Modulus values are non-dimensionalized ($K_h = \bar{K}_h/\bar{P}\bar{L}$).
(TIF)

## Acknowledgments

M.W. gratefully acknowledges the Center for Computational Biology at the Flatiron Institute for hosting her sabbatical, during which a substantial portion of this work was completed. We would like to thank L. Ramondo for obtaining a subset of the experimental data, and D. Albrecht and V. Kamara for production and assistance of our microfluidic device set-up. We would like to thank N. Olaranont for advice during the revision process.

## Author contributions

**Conceptualization:** Rholee Xu, Luis Vidali, Min Wu.

**Data curation:** Rholee Xu.

**Formal analysis:** Rholee Xu, Min Wu.

**Funding acquisition:** Luis Vidali, Min Wu.

**Investigation:** Rholee Xu.

**Methodology:** Rholee Xu, Luis Vidali, Min Wu.

**Project administration:** Luis Vidali, Min Wu.

**Resources:** Luis Vidali, Min Wu.

**Software:** Rholee Xu, Min Wu.

**Supervision:** Luis Vidali, Min Wu.

**Validation:** Rholee Xu, Min Wu.

**Visualization:** Rholee Xu.

**Writing – original draft:** Rholee Xu.

**Writing – review & editing:** Rholee Xu, Luis Vidali, Min Wu.

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
