## [Decision Letter · Decision Letter 0]

12 Nov 2025

PCOMPBIOL-D-25-01862

A surface morphology-based inference method for the cell wall elasticity profile in tip-growing cells

PLOS Computational Biology

Dear Dr. Xu,

Thank you for submitting your manuscript to PLOS Computational Biology. After careful consideration, we feel that it has merit but does not fully meet PLOS Computational Biology's publication criteria as it currently stands. In addition to several detailed comments, the two reviewers found that (1) the writing and logic was difficult to follow, and (2) point to a lack of comparison to experimental data. Therefore, we invite you to submit a revised version of the manuscript that addresses the points raised during the review process.

When preparing your revision and response, please also ensure compliance with the PLOS Computational Biology Code Availability policy (https://journals.plos.org/ploscompbiol/s/code-availability) in addressing comments regarding source code availability.

We look forward to receiving your revised manuscript.

Kind regards,

Dimitrios Vavylonis

Section Editor

PLOS Computational Biology

**Journal Requirements:**

**Reviewers' comments:**

Reviewer's Responses to Questions

**Comments to the Authors:**

Reviewer #1: "A surface morphology-based inference method for the cell wall elasticity profile in tip-growing cells " by Xu et al.

The authors present a protocol to compute the elastic strains and material properties of the wall of a tip-growing cell based on 3D stacks acquired from fluorescently marked cells before and after plasmolysis. The protocol is followed by an extensive analysis of the robustness of the estimates using simulated data. The paper offers many interesting observations but is rather difficult to follow. I would like to make a few suggestions to improve the clarity.

1. I don't understand the overall logic of the protocol. Right from the start (i.e. in the abstract), the authors state that "Previous work is based on the wall meridional outline on the assumption the cell is axisymmetric, an idea that does not align well with the reality of the cells.". The observed deviation from perfect axial symmetry is the motivation for the new experimental protocol based on image stacks and the positioning of cell surface markers in 3D without assuming axial symmetry. The puzzling part, for me, is that once the curvature and strains are computed on a 3D triangulated surface, these variables are then projected onto an axisymmetric shape (Fig. 4) and they are ultimately averaged between several cells to reduce noise (Fig. 6). The proposed method is thus technically challenging, because it starts with the localization of markers in 3D space; and yet it concludes with a "projection" of the variables on an axisymmetric surface. In my opinion, the authors take a more challenging path to arrive at the same endpoint as all prior protocols. Moreover, inspection of the 3D reconstructions of an actual cell in Fig. 2F and G would suggest that these reconstructions are too noisy to be useful on their own. The projection onto an axisymmetric surface seems almost necessary for a quantitative interpretation of the cell surface curvature and strain. So, why not start with the axisymmetric geometry?

From a biophysical standpoint, a 3D reconstruction of the cell surface is most useful if the cells under study deviate SYSTEMATICALLY from axial symmetry. That would be the case for cells that are growing along a well-defined 2D curve or some 3D helical path. While some tip-growing cells do show these behaviors, most tip-growing cells are surprisingly good at maintaining their growth direction and shape. As a result, the small deviations from axial symmetry tend to cancel out if a cell is tracked for some time or if many cells are compared among themselves. Without clear, repeatable, deviations from axial symmetry, it is not obvious what is the benefit of a 3D reconstruction.

Given the observations stated above, I'm confused as to why the protocol presented is superior to measuring the cell curvature and strain directly on a median section of the cell. Experimentally and computationally, the standard protocol is easier to implement and potentially more robust in its estimate of the curvature and strain because it relies on the x,y-position of the cell contour and surface markers, and leaves out the z position whose spatial resolution is far inferior to the x,y-position in most microscope setups. I'm thinking especially of the pole of the cell where a robust 3D reconstruction is particularly challenging when the cell is observed on its side. I would therefore ask the following question to the authors:

What are the concrete ways in which their protocol is superior to earlier protocols where axial symmetry is assumed from the start?

Finally, I should note that the lack of a perfect axial symmetry has been taken into account in some prior work. This was done by computing the curvature and other variables independently on the "left side" and "right side" of the cell. In all the cases that I can readily remember, the slight asymmetries in cell shape have little impact of the value of the variables that are computed.

2. On a related note, I would urge the authors to perform a direct quantitative comparison between their protocol and one based on the median section of the cell. It seems the authors already have the data to do a comparative study for the meridional curvature. As far as I can tell, Fig. S3 was prepared by extracting the contour from median sections of several cells. These data could be compared to the curvature that it arrived at by projection of 3D reconstructions. I would suggest that the authors do the same for the elastic strains. These comparisons would go a long way in justifying the use of the new protocol.

Also, the addition of more biological data would increase the appeal of the paper. As it stands, the focus of the manuscript is on simulated data instead of data from real cells.

3. I also several questions regarding the data acquisition protocol described at the beginning of the Methods section.

3.1. I believe the cell wall is fluorescently labeled with calcofluor and this information was used to make the reconstruction of Fig. 2F. It is not clear, however, whether the beads that are used to track the elastic strains are positioned in 3D such that they sit on the surface defined by the cell wall. Stated differently, a 3D reconstruction can be achieved by connecting the fluorescent beads into a triangulated surface (as seen in Fig. 2A) or by stacking the fluorescent wall outlines as in Fig. 2F. My question is whether these two sources of 3D information are forced to be compatible with each other. The "wall contour" approach is obviously the richest in terms of positional information. It would make sense to confined the beads to an interpolated surface defined by the wall outlines.

3.2. I'm confused with one aspect of the stereographic projection. I understand the principle of the Delaunay triangulation and the need for a well-defined 2D surface for the triangulation to be computed. If the interpolated surface mentioned above was computed, then a Delaunay triangulation could be computed on it. Instead, the authors chose a mixed stereographic projection of the cell surface markers onto a plane to implement the triangulation. The topology obtained on the plane is then used to triangulate the cell surface. I would like to know if the nice geometrical properties of the Delaunay triangulation are preserved when the triangles are mapped back onto the cell surface. I don't remember clearly all the properties of a proper Delaunay triangulation but it seems that one of them is that the triangles are made as "equilateral" as possible given the set of points. My concern is that the large geometrical deformation imposed by the stereographic projection could lead to a triangulation that is not optimal once it is applied to the curved cell surface. Did the authors investigate that? Would it not be possible to perform the Delaunay triangulation directly on the interpolated cell surface obtained from the calcofluor staining?

3.3. A final point, it is not clear how the equivalent points on the turgid and plasmolyzed cell are matched with each other. It can be a very tedious task, especially if the number of markers is large and the change in cell geometry is substantial. Was the matching done manually?

4. Finally, the text should be checked thoroughly. There are many sentences that are poorly phrased, obscure, or plainly wrong. I'm listing some issues below but I must confess that I gave up after a while.

try to eliminate or shorten the "noun chains" that appear in the manuscript. For example, the expressions "Marker point triangulation scheme", "fluorescent bead covered surface", "cell wall surface marker points" are hard to read.

Abstract

"Plant cell morphology and growth are essential for plant development and adaptation, in particular the cell wall material deposition and rearrangement. " This sentence is awkward. I would rephrase it.

"As the wall is extended due to turgor pressure, the wall mechanical response, specifically the wall’s elastic properties, are not well understood. " This sentence is nonsensical.

"Previous work is based on the wall meridional outline on the assumption the cell is axisymmetric, an idea that does not align well with the reality of the cells. " Please clarify what you mean.

"Instead, we developed a way to triangulate the surface of the cell using marker point locations that could be achieved experimentally with fluorescent labeling. " Rephrase.

"a mechanical model is used to measure the tensions and surface bulk modulus distribution " Models are not used to "measure".

"We simulated moss tip cells from experimental Physcomitrium patens plants to challenge the method with noise estimated from experiments. " This sentence is nonsensical.

"causing false-positive gradient results " Please clarify what you mean.

"understanding tip cell growth " The expression "tip cell" appears in many places. It is not clear if you are talking about the "cell's tip" or a "tip-growing cell".

Author summary

"Their growth and morphogenesis are tightly regulated processes involving cell wall addition and rearrangement, which aim to reduce the wall stress originating from the cell’s internal turgor pressure. " wall addition and rearrangement is not "aimed" at reducing the wall stresses.

"We simplify this cell growth system by first studying the cell wall’s elastic properties that are present without active growth. " This sentence is awkward. Moreover, the elasticity of the cell wall is present whether or not growth is occurring. Your own "Dual-Configuration Model" says so.

"We use a surface triangulation method derived from marker point tracking to measure the spatial elasticity along the tip cell " Rephrase.

"Our results suggest that this inference method can reliably measure a cell wall elasticity gradient under combined geometric and mechanical conditions that create large enough elastic strain at the tip. " Clarify "large enough". 5%? 10%? 50%?

line 7 "those mechanics differ in particular areas along the cell surface. " "those mechanics"? Do you mean "mechanical properties"?

line 19 "The scale of single tip-growing cells make it difficult to measure wall mechanics, especially at discrete spatial locations. " SMALL scale (give the size) makeS. What do you mean with "measure wall mechanics". Do you mean to "measure the wall's mechanical properties"

line 22 "In tip-growing cells, the use of micro-indentation has been a common tool used to measure local mechanical differences in pollen tubes [13–15]. Due to the geometrical orientation of the probing, the measurement may not provide a direct measurement of the mechanics relevant to wall expansion " The first sentence should be rephrase. Also both sentence have awkward repetitions of words: use(d) and measurement.

line30 "The models note that a steep gradient of mechanical properties ..." "The models note ..."? I would say "Results from the models indicate ..."

line 32 "An alternative to Lockhart is the idea that the wall growth comes from the combination of two actions, the first ... reversible elastic deformation ... the second action involved is the irreversible wall expansion" The view proposed is NOT an alternative to Lockhart because the very first equation in Lockhart's paper (Eq. 1) is a statement that the observed change in cell length is the sum of reversible and irreversible deformation of the wall. The reason why wall elasticity does not appear in the Lockhart Equation is because it is the relation for steady-state growth, i.e. when pressure is not changing in time.

line 43 "Importantly, the results showed a gradient of properties within one order of magnitude. " What properties? What gradient?

line 63 "cell wall surface marker points " how about "cell surface markers"

line 64, the six lines starting at lines 64 include details that are specific to the methods of this paper, which is the purpose of the methods section, and methodological details that are not used in this paper. "In experiments, the marker points can come from the adhesion of fluorescent beads or 65 quantum dots, for instance [9,20,23] ". "The unturgid state can be obtained by removing turgor pressure 68 with an osmoticum change [23,29] (Fig 1A bottom) or laser ablation [27,30]." I would eliminate from the methods section the mentions of alternative protocols and focused instead on explaining clearly what was done.

line 117 "The standard calculation of principal curvatures in the outline model and previous works is denoted by: κ $\bar{s}$ = dα/ds and κ $\bar{\theta}$ = sin α/r ". I don't understand

line 119 "On the wall surface, the s and r parameters are hard to define even in the optimal sense. " Clarify.

line 157 "In the axisymmetric system, the wall properties can be defined on two orthogonal directions along the circumferential (*θ*) and meridional (s) coordinates. " On an axisymmetric geometry, all variables can be expressed in terms of arc length position (s) because there is no variation in the circumferential direction.) and meridional (s) coordinates. " On an axisymmetric geometry, all variables can be expressed in terms of arc length position (s) because there is no variation in the circumferential direction.) and meridional (s) coordinates. " On an axisymmetric geometry, all variables can be expressed in terms of arc length position (s) because there is no variation in the circumferential direction.) and meridional (s) coordinates. " On an axisymmetric geometry, all variables can be expressed in terms of arc length position (s) because there is no variation in the circumferential direction.

line 170 "For simplicity, we assume the 2D nonlinear Hookean constitutive law relates the tensions with the elastic stretch ratios through two material properties, the surface bulk and shear modulus, K $\bar{h}$ and *μ* $\bar{h}$, respectively" Rephrase

lline 181 "Subsequently, K $\bar{h}$ and *μ* $\bar{h}$ can be be measured by rearranging Eqs 7 and 8: " Parameters are not "measured" by rearranging equations.

line 193 "Due to the isotropic nature of the tip cell, this will always occur at the tip." Here, I assume "tip cell" means "pole"

Figure 2 and Figure 5, the colormaps are almost opposite. In Fig. 2, red is low and blue is high. In Fig. 5, blue is low and yellow is high.

line 522 "The spatial cell wall elasticity in tip-growing cells is one determining factor of where plant cell walls grow, making them an important measurement for tip growth models. " What is "them" refering to?

line 525 "By using a surface triangulation method to connect marker points across the cell surface, we were able to adopt the axisymmetric mechanical model to measure the wall elastic moduli (Fig 1, 2). " do you mean "adapt"?

line 570 "that a courser spacing of marker points " Do you mean "coarser"?

Finally, the movies in supplementary material do not seem very useful. Static plots would be more easily compared.

Reviewer #2: Dear authors,

Please find my review in the attached pdf.

**Have the authors made all data and (if applicable) computational code underlying the findings in their manuscript fully available?**

The PLOS Data policy requires authors to make all data and code underlying the findings described in their manuscript fully available without restriction, with rare exception (please refer to the Data Availability Statement in the manuscript PDF file). The data and code should be provided as part of the manuscript or its supporting information, or deposited to a public repository. For example, in addition to summary statistics, the data points behind means, medians and variance measures should be available. If there are restrictions on publicly sharing data or code —e.g. participant privacy or use of data from a third party—those must be specified.requires authors to make all data and code underlying the findings described in their manuscript fully available without restriction, with rare exception (please refer to the Data Availability Statement in the manuscript PDF file). The data and code should be provided as part of the manuscript or its supporting information, or deposited to a public repository. For example, in addition to summary statistics, the data points behind means, medians and variance measures should be available. If there are restrictions on publicly sharing data or code —e.g. participant privacy or use of data from a third party—those must be specified.requires authors to make all data and code underlying the findings described in their manuscript fully available without restriction, with rare exception (please refer to the Data Availability Statement in the manuscript PDF file). The data and code should be provided as part of the manuscript or its supporting information, or deposited to a public repository. For example, in addition to summary statistics, the data points behind means, medians and variance measures should be available. If there are restrictions on publicly sharing data or code —e.g. participant privacy or use of data from a third party—those must be specified.requires authors to make all data and code underlying the findings described in their manuscript fully available without restriction, with rare exception (please refer to the Data Availability Statement in the manuscript PDF file). The data and code should be provided as part of the manuscript or its supporting information, or deposited to a public repository. For example, in addition to summary statistics, the data points behind means, medians and variance measures should be available. If there are restrictions on publicly sharing data or code —e.g. participant privacy or use of data from a third party—those must be specified.

Reviewer #1: Yes

Reviewer #2: **No:**

PLOS authors have the option to publish the peer review history of their article (what does this mean?). If published, this will include your full peer review and any attached files.). If published, this will include your full peer review and any attached files.). If published, this will include your full peer review and any attached files.). If published, this will include your full peer review and any attached files.

...

Reviewer #1: No

Reviewer #2: **Yes:** Olivier AliOlivier AliOlivier AliOlivier Ali

**Figure resubmission:**
---

## [Decision Letter · Decision Letter 1]

16 Mar 2026

PCOMPBIOL-D-25-01862R1

A surface morphology-based inference method for the cell wall elasticity profile in tip-growing cells

PLOS Computational Biology

Dear Dr. Xu,

Thank you for submitting your manuscript to PLOS Computational Biology. After careful consideration, we feel that it has merit but does not fully meet PLOS Computational Biology's publication criteria as it currently stands. Therefore, we invite you to submit a revised version of the manuscript that addresses the points raised during the review process.

We look forward to receiving your revised manuscript.

Kind regards,

Dimitrios Vavylonis

Section Editor

PLOS Computational Biology

**Reviewers' comments:**

Reviewer's Responses to Questions

**Comments to the Authors:**

Reviewer #1: I thank the authors for taking the time to answer my comments. I am satisfied with their answers. I have a final comment regarding the new Fig. 6. The stresses (Fig. 6D, second row) should be in the ratio 2:1 (circumferential:meridional) at the base of the tip (local angle = 90 deg.). For both cell types (especially the chloronema), the experimental data deviate from this ratio. Could the authors explain why? I could not found an explanation in the main text.

Reviewer #2: Dear Authors,

Thank you for this revisited version of your manuscript and the significative amount of work and effort you put into it.

It seems that most of my main initial concerns have been addressed and I am please to support the acceptance of your manuscript.

I however suggest two (very) minors (and optional) upgrades :

1. Maybe I missed it, but I think the github repository is not explicitly mentioned in the main text. It would be worth having a "Code availability" subsection, at the end of the "Methods" section, pointing directly to this repository. Or maybe this is something handled by the editorial team directly ?

2. Concerning the code repository, it seems to serve its purpose and the detailed README file is appreciated. However, a more throughout description of the installation procedure mentionning the required softwares and where to get them (inserted at the beginning of this README) would certainly help the interested readers to use the proposed code.

Best regards,

**Have the authors made all data and (if applicable) computational code underlying the findings in their manuscript fully available?**

The PLOS Data policy requires authors to make all data and code underlying the findings described in their manuscript fully available without restriction, with rare exception (please refer to the Data Availability Statement in the manuscript PDF file). The data and code should be provided as part of the manuscript or its supporting information, or deposited to a public repository. For example, in addition to summary statistics, the data points behind means, medians and variance measures should be available. If there are restrictions on publicly sharing data or code —e.g. participant privacy or use of data from a third party—those must be specified.requires authors to make all data and code underlying the findings described in their manuscript fully available without restriction, with rare exception (please refer to the Data Availability Statement in the manuscript PDF file). The data and code should be provided as part of the manuscript or its supporting information, or deposited to a public repository. For example, in addition to summary statistics, the data points behind means, medians and variance measures should be available. If there are restrictions on publicly sharing data or code —e.g. participant privacy or use of data from a third party—those must be specified.requires authors to make all data and code underlying the findings described in their manuscript fully available without restriction, with rare exception (please refer to the Data Availability Statement in the manuscript PDF file). The data and code should be provided as part of the manuscript or its supporting information, or deposited to a public repository. For example, in addition to summary statistics, the data points behind means, medians and variance measures should be available. If there are restrictions on publicly sharing data or code —e.g. participant privacy or use of data from a third party—those must be specified.requires authors to make all data and code underlying the findings described in their manuscript fully available without restriction, with rare exception (please refer to the Data Availability Statement in the manuscript PDF file). The data and code should be provided as part of the manuscript or its supporting information, or deposited to a public repository. For example, in addition to summary statistics, the data points behind means, medians and variance measures should be available. If there are restrictions on publicly sharing data or code —e.g. participant privacy or use of data from a third party—those must be specified.

Reviewer #1: Yes

Reviewer #2: Yes

PLOS authors have the option to publish the peer review history of their article (what does this mean?). If published, this will include your full peer review and any attached files.). If published, this will include your full peer review and any attached files.). If published, this will include your full peer review and any attached files.). If published, this will include your full peer review and any attached files.

...

Reviewer #1: **Yes:** Jacques DumaisJacques DumaisJacques DumaisJacques Dumais

Reviewer #2: **Yes:** Olivier ALIOlivier ALIOlivier ALIOlivier ALI

**Figure resubmission:**
---

## [Editor Report · Decision Letter 2]

22 Mar 2026

Dear Ms. Xu,

We are pleased to inform you that your manuscript 'A surface morphology-based inference method for the cell wall elasticity profile in tip-growing cells' has been provisionally accepted for publication in PLOS Computational Biology.

Best regards,

Dimitrios Vavylonis

Section Editor

PLOS Computational Biology

---

## [Editor Report · Acceptance letter]

PCOMPBIOL-D-25-01862R2

A surface morphology-based inference method for the cell wall elasticity profile in tip-growing cells

Dear Dr Xu,

I am pleased to inform you that your manuscript has been formally accepted for publication in PLOS Computational Biology. Your manuscript is now with our production department and you will be notified of the publication date in due course.

With kind regards,

Anita Estes
